# Phylogenetically and functionally diverse microorganisms reside under the Ross Ice Shelf

Clara Martínez-Pérez [1,2,17], Chris Greening [3,4], Sean K. Bay [3,4], Rachael J. Lappan [3], Zihao Zhao [1], Daniele De Corte[5], Christina Hulbe [6], Christian Ohneiser [7], Craig Stevens [8,9], Blair Thomson[10], Ramunas Stepanauskas [11], José M. González [12], Ramiro Logares [13], Gerhard J. Herndl [1,14,15], Sergio E. Morales [16 ✉] & Federico Baltar [1,10 ✉]

Throughout coastal Antarctica, ice shelves separate oceanic waters from sunlight by hundreds of meters of ice. Historical studies have detected activity of nitrifying microorganisms in oceanic cavities below permanent ice shelves. However, little is known about the microbial composition and pathways that mediate these activities. In this study, we profiled the microbial communities beneath the Ross Ice Shelf using a multi-omics approach. Overall, beneath-shelf microorganisms are of comparable abundance and diversity, though distinct composition, relative to those in the open meso- and bathypelagic ocean. Production of new organic carbon is likely driven by aerobic lithoautotrophic archaea and bacteria that can use ammonium, nitrite, and sulfur compounds as electron donors. Also enriched were aerobic organoheterotrophic bacteria capable of degrading complex organic carbon substrates, likely derived from in situ fixed carbon and potentially refractory organic matter laterally advected by the below-shelf waters. Altogether, these findings uncover a taxonomically distinct microbial community potentially adapted to a highly oligotrophic marine environment and suggest that ocean cavity waters are primarily chemosynthetically-driven systems.

[1] Department of Functional and Evolutionary Ecology, University of Vienna, Djerassiplatz 1, 1030 Vienna, Austria. [2] Centre for Microbiology and Environmental Systems Science, Division of Microbial Ecology, University of Vienna, Djerassiplatz 1, 1030 Vienna, Austria. [3] Department of Microbiology, Biomedicine Discovery Institute, Monash University, Clayton, VIC 3800, Australia. [4] Securing Antarctica's Environmental Future, Monash University, Clayton, VIC 3800, Australia. [5] Institute for Chemistry and Biology of the Marine Environment, Carl von Ossietzky University of Oldenburg, Oldenburg, Germany. [6] School of Surveying, University of Otago, Dunedin, New Zealand. [7] Department of Geology, University of Otago, Dunedin, New Zealand. [8] National Institute of Water and Atmospheric Research, Greta Point, Wellington 6021, New Zealand. [9] Department of Physics, University of Auckland, Auckland, New Zealand. [10] Department of Marine Sciences, University of Otago, Dunedin, New Zealand. [11] Bigelow Laboratory for Ocean Sciences, East Boothbay, ME, USA. [12] Department of Microbiology, University of La Laguna, ES-38200 La Laguna, Spain. [13] Department of Marine Biology and Oceanography, Institut de Ciències del Mar (CSIC), Barcelona, Spain. [14] NIOZ, Department of Marine Microbiology and Biogeochemistry, Royal Netherlands Institute for Sea Research, Utrecht University, PO Box 59, 1790 AB Den Burg, The Netherlands. [15] Vienna Metabolomics Center, University of Vienna, Djerassiplatz 1, A-1030 Vienna, Austria. [16] Department of Microbiology and Immunology, University of Otago, Dunedin, New Zealand. [17] Present address: Institute for Environmental Engineering, Department of Civil, Environmental and Geomatic Engineering, Eidgenössische Technische Hochschule (ETH) Zürich, 8093 Zurich, Switzerland. ✉email: sergio.morales@otago.ac.nz; federico.baltar@univie.ac.at

Ice shelves are permanent floating extensions of grounded sheets of ice that connect to a landmass. The Ross Ice Shelf, by area the largest ice shelf in the world, floats atop an ~54,000 km³ ocean cavity that covers about half of the Ross Sea and hugs the coast of Antarctica (Fig. 1a). Generally over 300 m thick[1], the ice shelf creates a "lid" that isolates the underlying ocean from the atmosphere and from sunlight, and exerts a direct effect on the chemical composition of the water column beneath it (in general ~700 m deep[2]). Waters under the permanent ice shelves are influenced by continental ice-sheet melting and are thus an important intermediary between subglacial outflow from the Antarctic continent and the open Ross Sea, and ultimately the Southern Ocean. Despite their oceanographic significance, sub-ice shelf habitats are among the least-studied ecosystems in the world's oceans.

Oceanographic and biogeochemical observations of the water cavity beneath the Ross Ice Shelf have been largely concentrated on the shelf margins, in particular at the McMurdo Ice Shelf (northwestern portion of the Ross Ice Shelf). Here, nutrient- and biomass-rich water advected from eastern McMurdo Sound likely plays an important role in sub-ice biogeochemistry of the dark ecosystem beneath the shelf front[3,4]. Direct observations in the grounding area have also confirmed a stratified and quiescent ocean setting[5]. As a result, water below the Ross Ice Shelf is reported to be exchanged with the Ross Sea with an estimated residence time of 0.9–5.4 years[6,7]. This allows transport of nutrients and organisms from the sea into the cavity. However, unlike other well-ventilated shelves (e.g., Amery shelf[8]), the proximity to open water is likely a major factor controlling biogeochemical process in the central basin of the Ross Ice Shelf cavity.

Opportunities to directly access the central sub-ice shelf cavity have been greatly limited by logistical constraints and only one expedition to date has sampled the seawater beneath the center of the Ross Ice Shelf. Sampling of the sub-ice water column took place through borehole J9, during the Ross Ice Shelf Project of 1977[9]. The environment beneath the Ross Ice Shelf was described as "similar to the abyssal ocean in being cold and aphotic". Within these waters, "sparse" populations of bacteria, microbial eukaryotes, and animals were observed[10,11]. The microbial populations were proven to be heterotrophically active and incorporated radiolabeled organic carbon molecules at very low rates comparable to the abyssal ocean[10]. Autotrophic activity of these microbial communities was subsequently reported and attributed to "nitrifying bacteria"[12]. In these aphotic ecosystems lacking photosynthetic primary production, dark carbon fixation by nitrifying microorganisms may be sufficient to sustain observed microbial and macrofaunal populations[12]. Lateral inputs of organic carbon from the Ross Sea may also support these populations. However, given these studies preceded the advent of molecular techniques, the composition of the microbial communities, their relatedness to open ocean communities, and their possible links to ecosystem function remained unexplored.

In this study we accessed the waters beneath the Ross Ice Shelf to uncover the phylogenetic and functional diversity of the microbial communities under the Antarctic ice shelf. We combined multi-omics techniques (metagenomics, metatranscriptomics, single-cell genomics) with supporting biogeochemical measurements (nutrient measurements and heterotrophic bacterial production). We show that the waters below the shelf harbor a diverse microbial community with a taxonomic composition distinct from other open ocean environments. In addition, we observed the transcription of various genes associated with lithoautotrophic and organoheterotrophic growth, uncovering the basis for previous activities reported in below-shelf waters.

## Results

**The water column under the Ross Ice Shelf is characterized by a steep vertical ammonium gradient.** During the Ross Ice Shelf Program in December 2017, an access borehole was created by hot water drilling at site HWD-2 (latitude 80.6577 S, longitude

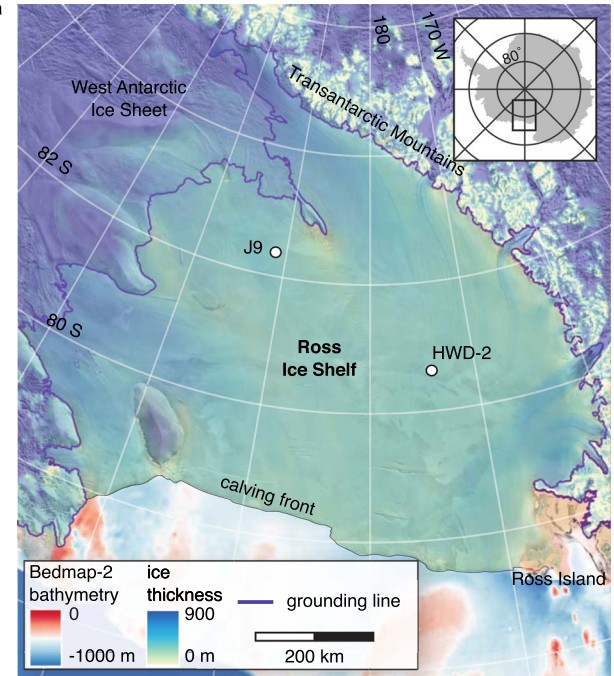

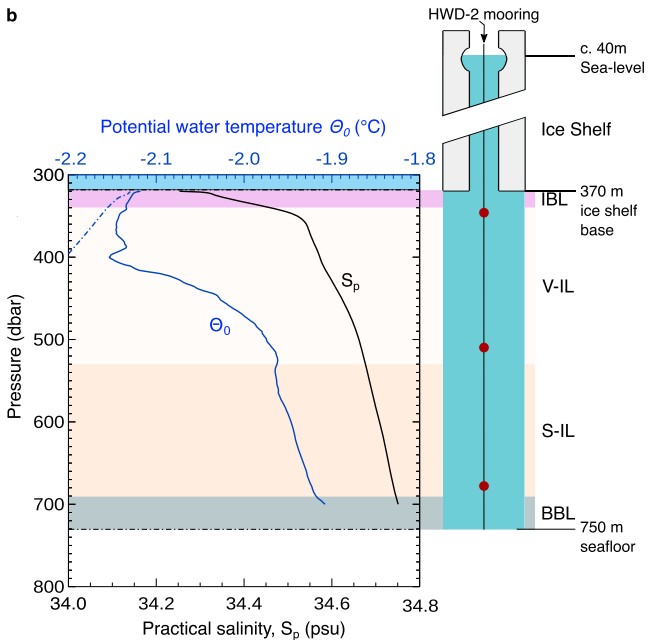

**Fig. 1 Sampling location. a** Map showing the sampling location of this study (HWD-2) and the borehole study site J9 drilled in 1977[10]. Bathymetry and ice thickness are based on the Bedmap-2 data set[1]. The transparent ice surface image was sourced from the MOA2009 image map[119]. **b** (left) Thermohaline structure of the water column at station HWD-2 and defined regions. IBL, Ice basal boundary layer. V-IL, variable intermediate layer, likely modulated by tides and resulting in patches of water with variable temperature and salinity. S-IL, stratified intermediate layer. BBL, benthic boundary layer. (right) Schematic of HWD-2 drilling site depicts the sampling location of seawater samples (red circles) at 30, 180, and 330 m below the ice shelf base.

**Table 1 Biogeochemical data from the water columns below the Ross Ice Shelf at the HWD-2 borehole.**

| Depth (m) | Temp (°C) | Practical salinity (psu) | $NH_3$ (μM) | $NO_x$ (μM) | $PO_4^{3-}$ (μM) | $SiO_2$ (μM) | Cell abundance (×$10^5$ cells ml$^{-1}$) | LNA (%) | HNA (%) | PHP (μmol C m$^{-3}$ d$^{-1}$) | Turnover time (d) |
|---|---|---|---|---|---|---|---|---|---|---|---|
| 30 | −2.13 | 34.57 | 0.44 (0.02) | 7.35 (0.23) | 0.720 (0.003) | 165.0 (2.1) | 0.9 (0.1) | 82.8 (0.2) | 18.0 (0.4) | 0.30 (0.02) | 461 |
| 180 | −1.96 | 34.69 | 0.05 (0.02) | 7.32 (0.04) | 0.720 (0.004) | 166.0 (0.7) | 1.20 (0.07) | 85.5 (0.3) | 15.1 (0.3) | 0.60 (0.03) | 339 |
| 330 | −1.91 | 34.76 | 0.04 (0.01) | 7.37 (0.05) | 0.710 (0.001) | 165.0 (0.3) | 0.80 (0.07) | 77.4 (0.9) | 23.4 (0.7) | 0.4 (0.1) | 386 |

Mean (standard deviation in parentheses) values are shown (n = 3).
*Temp* temperature, *LNA* low nucleic acid content cells, *HNA* high nucleic acid content cells, *PHP* prokaryotic heterotrophic production.

174.4626 W), approximately 300 km from the Ross Sea and 330 km northwest of borehole J9 (Fig. 1a). The shelf ice was 370 m thick, and the underlying waters extended to 750 m below the shelf surface (Fig. 1b). Triplicate samples were collected at three depths: 30, 180, and 330 m below the bottom of the shelf (i.e., the ice-water interface). These depths correspond to three regions based on the thermohaline structure of the water column: a basal boundary layer just beneath the ice (IBL), the upper part of an intermediate layer characterized by highly variable temperature and salinity (V-IL), and the lower part of the intermediate layer characterized by linear stratification (S-IL). A homogeneous benthic layer was observed (BBL) but not sampled (Fig. 1b; see[13] for a detailed physical oceanographic description of the study site). This structure confirmed that the cavity is filled southward by thermohaline convection in which dense, high salinity shelf water (HSSW) evolves into very cold (~−2 °C) but relatively fresh Ice Shelf Water (ISW). The temperature and salinity conditions suggest that, other than the boundary layer regions, water properties conform to Deep Ice Shelf Water, a mixture of high and low salinity shelf water and Antarctic Surface Water (AASW)[13]. Contrary to what previous studies detected at the shelf front[3,4], other regional water masses were not present at borehole HWD2. The flow of waters beneath the drilling site was 2 cm s$^{-1}$ towards the open ocean, suggesting a residence time for these waters of ca. 4 years[13]. This estimate is within the range of 1–6 years from previous ocean measurements[6] and modeling studies[2,14].

Nutrient concentrations beneath the center of the Ross Ice Shelf were generally lower than those measured at the edge of the of the ice shelf[3,4] and in deep waters of the Ross Sea[15]. Concentrations of $SiO_2$ (165–166 μM), $NO_x$ (7.32–7.37 μM) and $PO_4^{3-}$ (0.71–0.72 μM) were relatively constant across the water column (Table 1) and two- to fourfold lower than in the oceanic cavity of the McMurdo Ice Shelf at the edge of the Ross Ice Shelf[3,4]. In contrast, we observed a steep gradient of ammonium, with concentrations tenfold higher at the basal layer (440 nM) than in deeper waters (40–50 nM). Such high ammonium concentrations, while lower than those in open waters of the Ross Sea (which peak in summer with values >2 μM;[15]), were in the same range as deep (400 m) high-salinity shelf waters (HSSW) entering the front of the cavity (~500 nm;[4]). A similar nutrient profile was reported beneath borehole J9[12], where ammonium concentrations were higher beneath the ice shelf base and decreased with depth, whereas values of $NO_3^-$ and $NO_2^-$ remained constant throughout the water column. However, concentrations of ammonium and $NO_x$ were 10- and 4-times higher at the J9 borehole than we reported for the HWD-2 borehole ($PO_4^{3-}$ and $SiO_2$ were not reported)[13,16,17].

Microbial cell abundance ranged from 0.9 to 1.2 × $10^5$ cells mL$^{-1}$ (Table 1), which is typical for mesopelagic and upper bathypelagic open ocean environments[18] and comparable to deep waters at the margin of the McMurdo Ice Shelf[4]. In contrast, prokaryotic heterotrophic production (PHP, a proxy for growth of heterotrophic organisms) ranged from 0.3 to 0.6 μmol C m$^{-3}$ d$^{-1}$ (Table 1), which is one to two orders of magnitude lower than at the margins of the Ross Ice Shelf (~40 μmol C m$^{-3}$ d$^{-1}$;[4]) and the average global PHP rates in the mesopelagic (24 μmol C m$^{-3}$ d$^{-1}$) and bathypelagic (4 μmol C m$^{-3}$ d$^{-1}$) open ocean[18]. Based on these PHP rates, the turnover time of the microbial community in our study ranged between 339 and 461 days, within the same order of magnitude as the approximately 400 days reported previously at borehole J9[10].

**Below-shelf microbial communities are distinct from open ocean communities.** Microbial community composition beneath the Ross Ice Shelf was determined using a combination of 16S

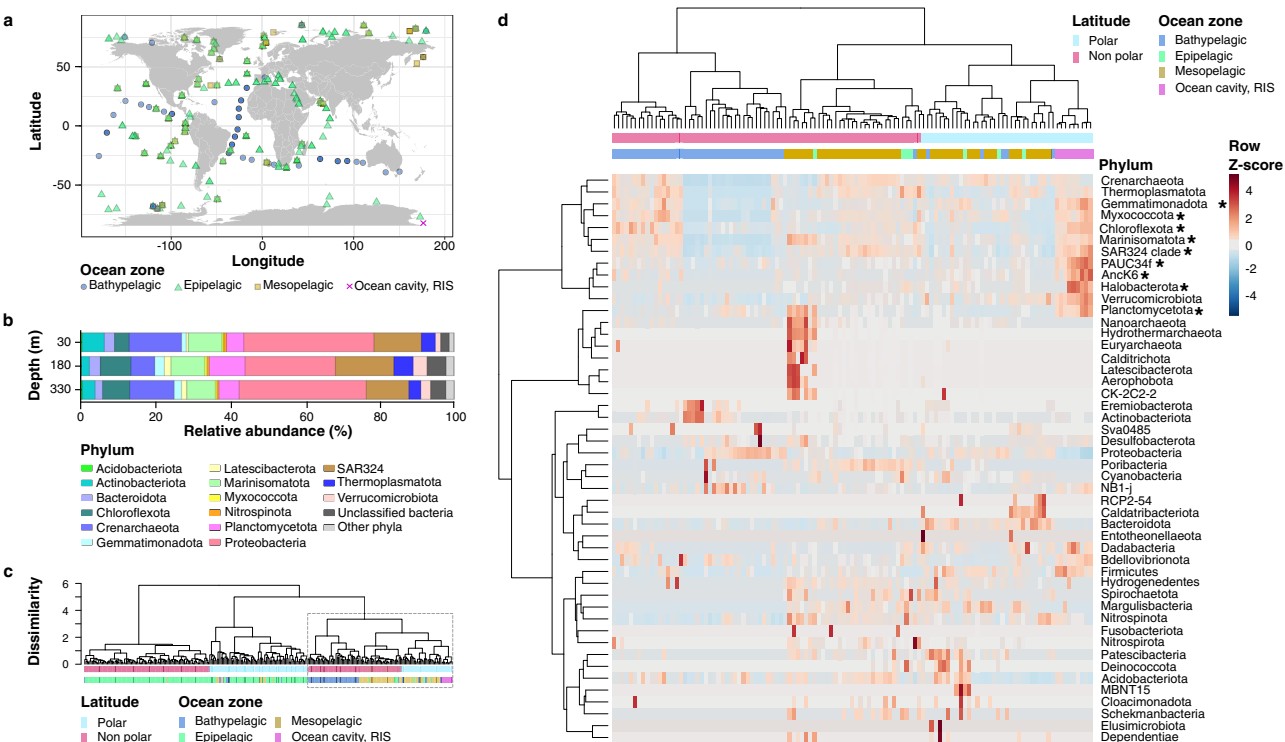

**Fig. 2 Comparison of bacterial and archaeal communities in the cavity beneath the Ross Ice Shelf with open ocean environments worldwide. a** Global map depicting the locations of metagenomic surveys utilized in the analysis and this study. Overlapping of symbols represent locations where multiple depths were sampled. **b** Phylum-level composition of microbial communities under the Ross Ice Shelf based on 16S rRNA amplicon sequencing (this study). The results for each sequencing triplicate are averaged; results for individual replicates and controls are shown in Supplementary Fig. 2a, b. Comparisons with metagenomic 16S ribosomal RNA genes (miTags) are shown in Supplementary Fig. 2c. **c** Cluster dendrogram depicting the average linkage hierarchical clustering based on a Bray-Curtis dissimilarity matrix of community compositions, based on the relative abundance of miTags from this study, global ocean expeditions, and Antarctic and Arctic surveys[20–23]. The dashed box highlights the clustering of communities in the ocean cavity under the Ross Ice Shelf with global deep-sea environments (in detail in **2d**). **d** Heatmap visualization of calculated Z-scores from below-shelf and global deep-sea environments, based on the relative abundance of miTags grouped at phylum level. Column dendrogram shows clustering of samples according to Bray-Curtis dissimilarity index (detailed from **2b**). Rows are clustered based on euclidean distance, grouping phyla that are most likely to co-occur in an environment. Asterisks mark phyla that are significantly more abundant under the Ross Ice Shelf (Kruskal-Wallis test, $p < 0.05$, Supplementary Data 3). Taxonomic assignment is based on the Genome Taxonomy Database (GTDB[107]).

rRNA gene amplicon sequencing and shotgun metagenomic sequencing. The microbial community was dominated by six phyla: Proteobacteria, SAR324, Crenarchaeota (mostly Nitrososphaerales), Marinisomatota (formerly Marinimicrobia, SAR406 clade), Chloroflexota (mostly SAR202), and Planctomycetota (Fig. 2b). Consistent with a dark oligotrophic environment, the eukaryotic community was largely comprised of taxa typically found in the meso- and bathypelagic open ocean, including Alveolata, Dinoflagellata, and Rhizaria lineages (Supplementary Fig. 1a, Supplementary Data 1). With respect to viruses, most bacteriophages detected in the metagenomic assemblies (~50%) belonged to uncultured or unclassified taxa (Supplementary Fig. 1b, Supplementary Data 2), with the most abundant classified viruses affiliating with the family Myoviridae (~30%).

We used 16S rRNA gene sequences extracted from metagenomic reads (miTags;[19]) to profile the relatedness of microbial communities beneath the Ross Ice Shelf to those of marine ecosystems globally (Fig. 2a, c[20–23],). This approach enabled comparison of microbial communities from available marine metagenomic datasets, while circumventing potential biases from inter-study community composition comparisons based on amplicon analyses[24]. In agreement with previous global metagenomic analyses[20], beta diversity analysis (Bray-Curtis dissimilarity) showed oceanic microbial communities cluster by depth,

though this was less pronounced in polar regions (Fig. 2c, d). In this global context, the communities beneath the Ross Ice Shelf form a cluster that is related to, but distinct from, those of mesopelagic polar open ocean waters (Fig. 2c, d). When compared to deep (>200 m) open ocean communities worldwide, compositional differences between open-ocean and below-shelf microbial communities are evident even at the phylum level (Fig. 2d). For example, the relative abundances of Chloroflexota, Gemmatimonadota, Marinisomatota, Myxococcota, Planctomycetota, and SAR324 were significantly higher under the Ross Ice Shelf, especially in deeper layers (Kruskal-Wallis test, $p = 9.4 \times 10^{-7} - 1.9 \times 10^{-5}$, full $p$ values shown in Supplementary Data 3). The phyla Halobacterota, Anck6 and PAUC34f, while typically rare in the open dark oceans, showed a tenfold increase in relative abundance in the cavity beneath the Ross Ice Shelf. Analyses restricted to polar environments using MGLM-ANOVA confirmed significant compositional differences between the ocean cavity and deep (>200 m) open-water polar environments (LRT = 17333, $p = 0.001$, Supplementary Data 3). In addition, Indicator Species Analysis (Indval) congruently identified 'signature species' of the ocean cavity (with respect to deep open-water polar communities) belonging to the phyla PAUC34f, Planctomycetota, and SAR324, as well as the classes Lentisphaeria, and SAR202 ($p = 0.001–0.002$, full $p$ values shown in

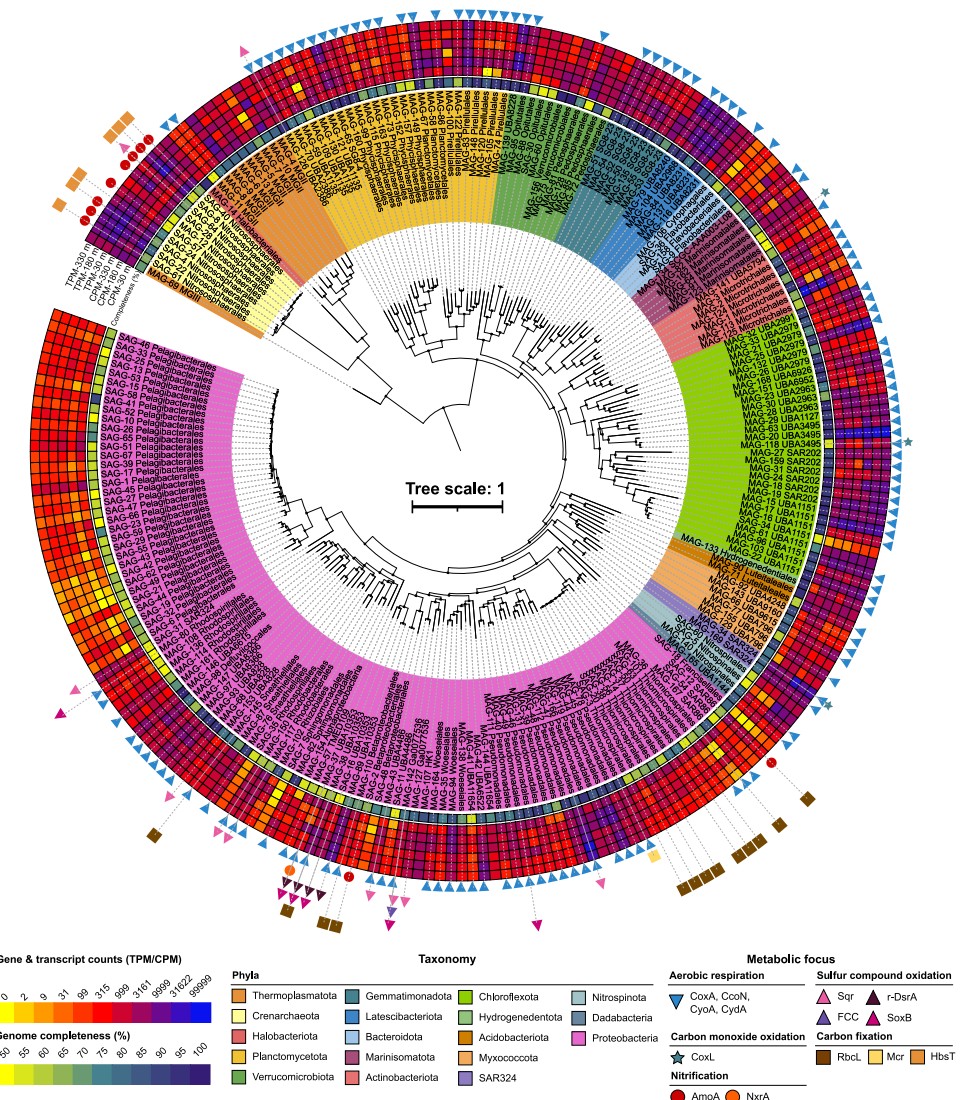

**Fig. 3 Phylogeny of reconstructed genomes under the Ross Ice Shelf.** Phylogenetic genome tree of the 235 metagenome-assembled genomes (MAGs) and single-amplified genomes (SAGs) retrieved from this study. The genomes are labeled by order, shaded by phylum, and numbered as per Supplementary Data 4. Genome characteristics (*inner-to-outer circular heatmap*): average genome completeness (%) at phylum level, relative abundance expressed as counts per million (CPM) and relative transcriptional activity as transcripts per million (TPM, Log10 + 1 transformed), and presence of marker genes for key metabolic pathways discussed in the main text.

Supplementary Data 3). These 'signature species' (with IndVal $p < 0.05$, test statistic >0.5, Supplementary Data 3) represented on average ~10% of the community beneath the Ross Ice Shelf, reaching up to 17% in the mid water column, in comparison to an average abundance of 0.75% in deep polar open waters.

Amplicon sequencing analysis provided additional taxonomic resolution of the communities under the ice shelf and confirmed the depth differentiation anticipated from oceanographic and chemical data. Significant differences in community alpha and beta diversity below the Ross Ice Shelf were observed between the basal boundary layer below the ice (30 m) and the deep water column (330 m) ($p = 0.028$, Supplementary Data 3, Supplementary Figs. 2 and 3). The species driving these differences are described in the Supplementary Notes.

**Nitrifying archaea and bacteria dominate transcription under the shelf.** We used a multi-omics approach to uncover the functional capacity of the microbial community beneath the Ross

Ice Shelf, integrating genome-resolved metagenomics, single-cell genomics, and metatranscriptomics. We assembled 235 dereplicated partial genomes (Fig. 3, Supplementary Figs. 4 and 5; Supplementary Data 4). These comprised 67 SAGs (single-amplified genomes) and 168 manually curated MAGs (metagenome-assembled genomes), all with completeness >50% and contamination <5%[25] (Fig. 3; Supplementary Data 4). These represent on average 50–60% of each sample's metagenomic and metatranscriptomic reads, including all phyla with relative abundance above 0.5% (Fig. 2) and the top four most abundant genera (Supplementary Fig. 2b). Their phylogenetic diversity, metabolic traits, and relative abundances are depicted in Fig. 3.

The presence and transcription of key metabolic genes in assembled and unassembled reads was used to identify prevailing metabolic pathways in the cavity under the Ross Ice Shelf. By far the most highly transcribed genes involved in autotrophic energy conservation pathways were those for oxidation of ammonium (ammonia monoxygenase, *amoA*) and nitrite (nitrite oxidoreductase, *nxrA*) (Fig. 4b). Accordingly, ammonium transporters and *amoA*

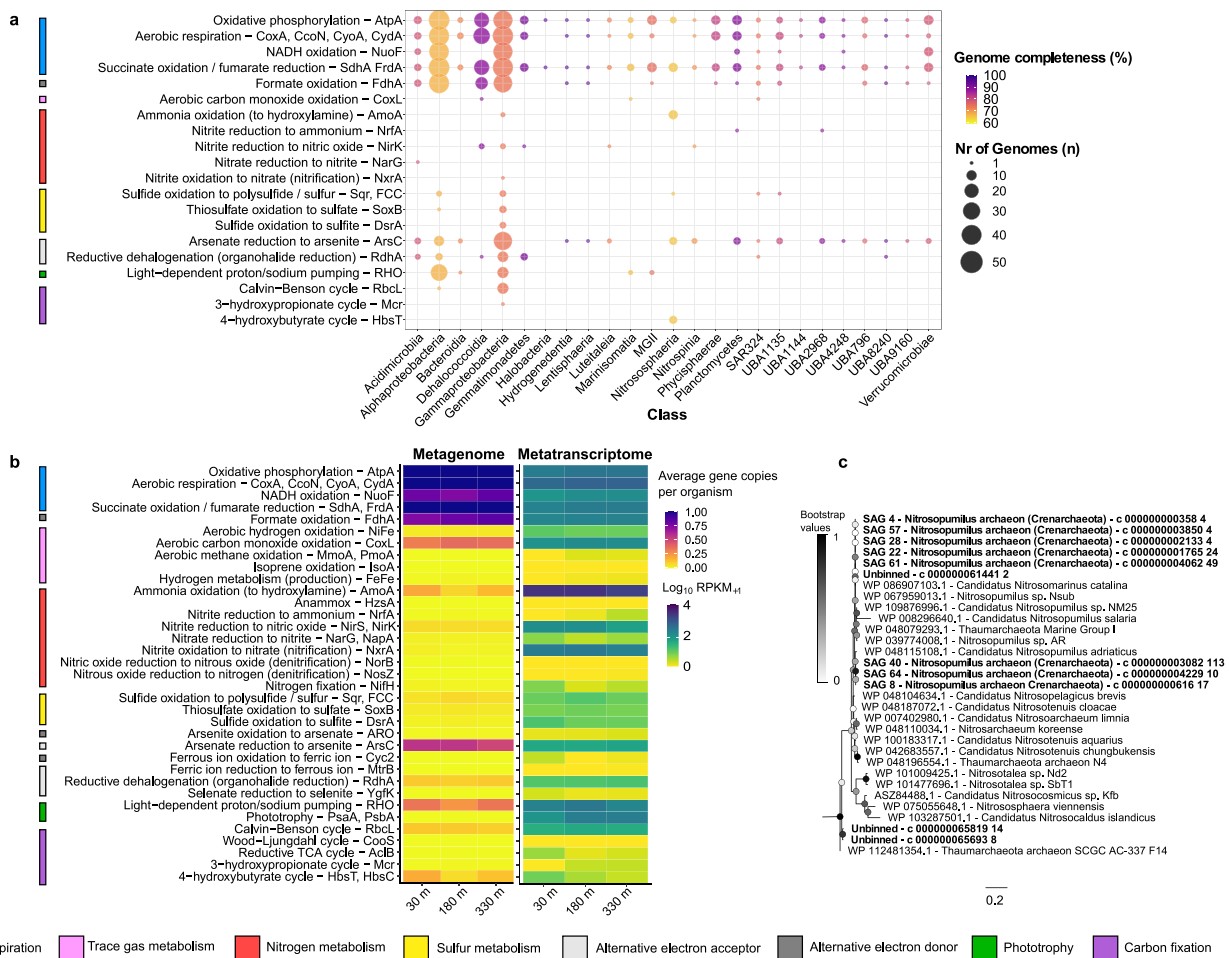

**Fig. 4 Energy conservation and carbon fixation strategies of communities beneath the Ross Ice Shelf. a** Dot plot showing the metabolic potential of the 235 metagenome-assembled genomes (MAGs) and single-amplified genomes (SAGs). The size class of each point represents the number of genomes in each class that encode the gene of interest and the shading represents the average genome completeness. **b** Heatmaps showing the relative abundance of these genes in the three metagenomic and metatranscriptomic unassembled short reads datasets. For metagenome reads, the heatmap shows the abundance of each pathway, expressed as average gene copies per organism (across all genes listed in the pathway) calculated relative to the abundance of 14 universal single-copy ribosomal genes, with scales capped at 1. For metatranscriptome reads, the heatmap shows log10-transformed reads per kilobase million (RPKM). Where genes within the same pathway are collapsed together, the values (community percentage or RPKM) are summed. **c** Phylogenetic tree of protein sequences of the highly transcribed ammonia monooxygenase subunit A (amoA) gene from archaeal single-amplified genomes and unbinned metagenomic contigs shown in bold compared to reference sequences. See Supplementary Fig. 7 for a detailed version of this tree.

were the most transcribed genes overall (Supplementary Fig. 6). Transcription patterns correlated with ammonium concentrations (Table 1) and relative abundance of the archaeal order Nitrosophaerales (Supplementary Figs. 2b, 4 and 5). Phylogenetic analysis corroborated that the most numerous amoA genes and transcripts were affiliated with Nitrosopumilus spp. (Fig. 4c, Supplementary Fig. 7), the most abundant and active archaeal lineage beneath the ice shelf (Supplementary Figs. 2b, 4 and 5), with some gammaproteobacterial amoA reads also detected (Fig. 4a, Supplementary Fig. 7). The metagenomic and metatranscriptomic reads of the marker gene for nitrite oxidation, nxrA, affiliated with the phyla Nitrospinota and, to a lesser extent Nitrospirota (Supplementary Data 5, Supplementary Fig. 8). In line with an autotrophic lifestyle, we identified the determinants of ammonium- or nitrite-dependent carbon fixation via the archaeal 4-hydroxybutyrate cycle (hbsC, hbsT genes) and Nitrospina reductive tricarboxylic acid cycle (aclB gene) (Fig. 4, Supplementary Figs. 9, 10 and 11; Supplementary Data 3).

Consistent with these results, reconstructed genomes from the genera Nitrosopumilus and Nitrospina were among those with highest relative transcriptional activity in our dataset (**S4, S5**).

These groups express a small fraction of their genomes (i.e., ~25% of total genes at 30 m) compared to other community members (Supplementary Fig. 4d–f), devoting most of their transcriptional effort to the key processes of carbon fixation and ammonia and nitrite oxidation, respectively. Despite being well-represented in the metatranscriptomic dataset, the relative abundance of the genus Nitrospina was low in the metagenomic dataset. For instance, the Nitrospina lineage represented by SAG_5 was among the least abundant genomes, but was highly active on the transcriptional level (RNA/DNA ~270; Supplementary Fig. 5) (Supplementary Data 4). These discrepant findings are in line with recent single-cell analyses showing Nitrospinota have high activity despite low abundance;[26] it is proposed that the large cell size or high mortality rates of these nitrite oxidizers are responsible for their low abundance in metagenomes and amplicon datasets compared to ammonium oxidizers[26,27].

**Various inorganic and organic energy sources likely support below-shelf bacteria.** Many members of the microbial community are capable of supporting or surviving beneath the shelf through a

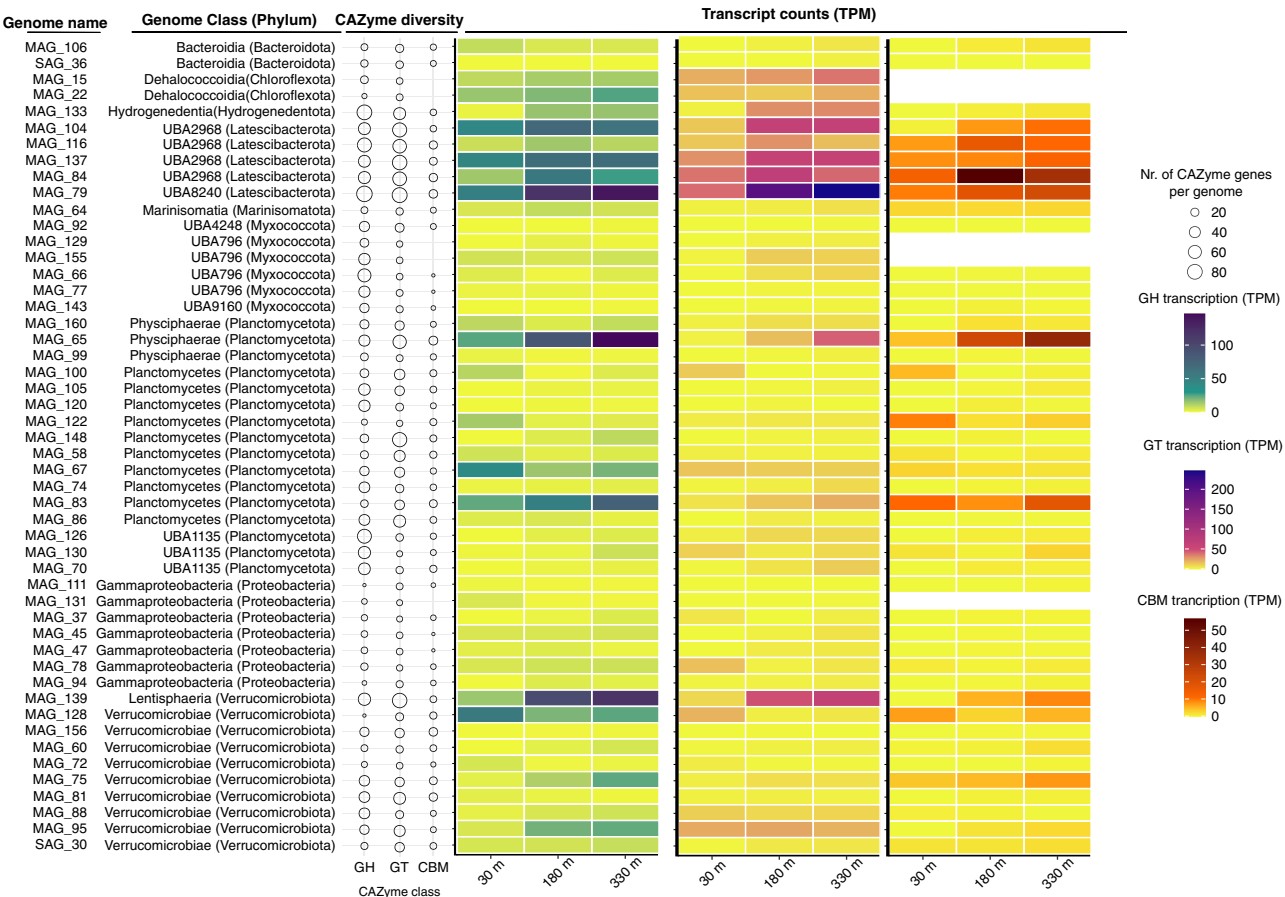

**Fig. 5 Relative abundance and transcription of selected carbohydrate active enzyme (CAZyme) classes.** Data is displayed for reconstructed genomes (MAGs and SAGs) where CAZyme diversity was highest (top 50 genomes). Bubble plots represent the number of different genes from each CAZyme class per genome (GH, glycosyl hydrolases; GT, glycosyl transferases; CBD, genes containing carbohydrate binding domains). Heatmaps represent the total gene transcription for each CAZyme class, normalized to total transcripts per sample (transcripts per million, TPM). The data used to construct these plots is provided in Supplementary Data 7.

chemoautotrophic or mixotrophic lifestyle. These include gamma-proteobacterial lineages, such as the Thioglobaceae (SUP05 and ARCTIC96BD-19) and UBA10353, which co-encode genes for the Calvin-Benson-Bassham cycle and heterotrophic metabolism. Consistently, RuBisCO genes ($rbcL$) affiliated to sulfur-oxidizing taxa (Supplementary Fig. 10) were transcribed at high levels throughout the water column (Supplementary Fig. 6). The potential of these lineages to fuel chemoautotrophy using reduced sulfur compounds as electron donors is supported by the presence and transcription of marker genes for sulfide oxidation ($sqr$, r-$dsrA$) and thiosulfate oxidation ($soxB$) (Fig. 4a, Supplementary Figs. 12, 13 and 14); (Supplementary Data 5 and 6). Abundant heterotrophic lineages, such as Marinisomatota and SAR324 (Fig. 2a, Supplementary Fig. 4), also encoded carbon monoxide dehydrogenases (Fig. 4a, Supplementary Fig. 15, Supplementary Data 6); carbon monoxide may serve as an energy source supporting persistence of this community, as we have recently described for other aerobic heterotrophic bacteria[28,29]. Genes for formate oxidation were also widespread and highly transcribed (Fig. 4b, Supplementary Fig. 6, Supplementary Data 6), whereas few community members are predicted to use $H_2$ (Supplementary Fig. 16, Supplementary Data 6).

Metabolic annotations of the derived genomes suggests that many identified taxa in this ecosystem adopt an organoheterotrophic lifestyle. Highly transcribed genes include a wide range of carbohydrate-active enzymes (CAZymes, Fig. 5,[30]), as well as the substrate-binding protein of the oligopeptide transporter (OppA; Supplementary Fig. 6). The highest enrichment (genes/Mbp),

diversity (number of different families), and transcripts of CAZymes were detected in reconstructed genomes of the phyla Hydrogenedentota, Latescibacterota, Myxococcota, Planctomycetota, and Verrucomicrobia. The CAZyme-rich genomes were among the most abundant (i.e., with highest coverage) in our study (Supplementary Fig. 4) and belong to the phyla enriched under the Ross Ice Shelf with respect to deep ocean environments (Fig. 2d). These genomes contained glycoside hydrolases, polysaccharide lyases, and glycosyltransferase families required for the utilization of heterogeneous polysaccharide chains, such as alginate, rhamnose, and xylan (Supplementary Data 7). These genomic features are consistent with previous studies describing the capability of these phyla to metabolize recalcitrant organic polymers[31–33]. Thus, the proportion of the community differentially enriched in this ecosystem could be adapted to degrade refractory organic compounds persisting in the advected waters beneath the Ross Ice Shelf. In contrast to their autotrophic counterparts, these heterotrophic populations transcribed a large percentage of their genome (~80%), especially in deeper waters (Supplementary Fig. 4d–f), with transcriptional effort spreading across a variety of substrate-utilization processes.

The metatranscriptome also revealed various other processes supporting life beneath the shelf. The heterotrophic majority in this system transcribed genes involved in the acquisition of inorganic and organic nitrogen and phosphorus compounds (e.g., urea, isocyanates, phosphonates, polyphosphonates; Supplementary Fig. 6). Genes encoding for cold adaptation processes (e.g.,

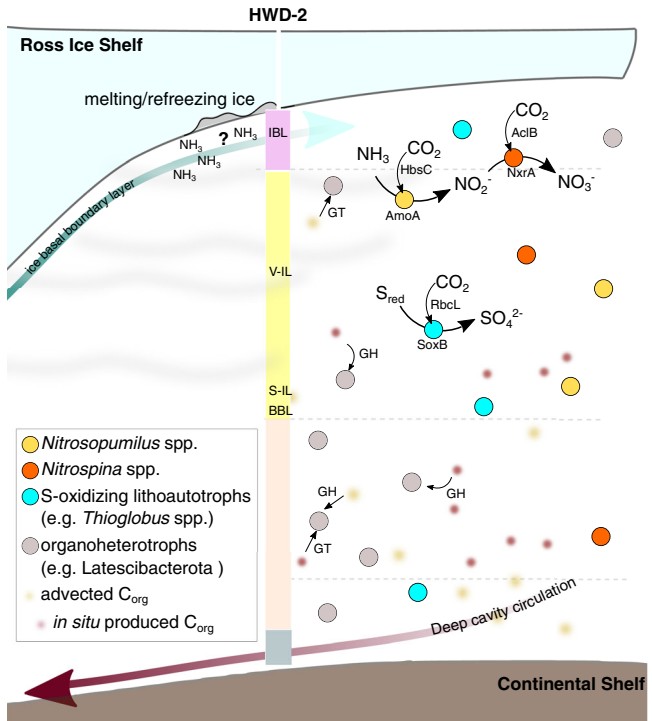

**Fig. 6 Schematic illustration of the dominant bacterial and archaeal groups in the water column under the Ross Ice Shelf.** Dotted lines represent the three depths sampled below the sea ice in this study (*not to scale; for a scaled representation, see* Fig. 1). At the lower fringe of the ice basal boundary layer (IBL), high concentrations of ammonium (from a yet unknown source) are likely to drive high relative abundance and transcriptional activity of ammonium oxidizing archaea (*Nitrosopumilus* ssp.) and nitrite oxidizing bacteria (*Nitrospina* ssp.). These, together with sulfur-oxidizing chemolithoautotrophs (belonging to e.g., the genus *Thioglobus*), are likely the main source of new organic matter to this ecosystem. The representative enzymes for the metabolic pathways are displayed only once for simplicity but were detected at all depths. The heterotrophic majority is characterized by metabolically versatile bacterial lineages (e.g., belonging to the phylum Latescibacterota), encoding and transcribing multiple copies of carbohydrate-active enzymes (CAZymes, such as glycosyl transferases GT, or glycosyl hydrolases, GH). These likely feed on in-situ generated or laterally advected complex organic matter.

cold-shock proteins), osmoregulation (e.g., glycine betaine transporters), and motility (i.e., flagellar apparatus) were highly transcribed (Supplementary Fig. 6). The constitutive expression of cold-shock chaperones can protect against cold-induced protein misfolding[34] and is likely an adaptive response to maintain protein homeostasis at the very low water temperatures below the shelf. Furthermore, transport of compatible solutes protects the cell against freezing, hyper-osmolality, and desiccation[35]. Glycine betaine transporters may provide an additional advantage given these transporters were recently shown to be multifunctional, as they transport multiple substrates in addition to the key osmoregulatory compound glycine betaine[36].

## Discussion

Collectively, our results provide a detailed insight on the ecological strategies adopted by communities living in the world's most extensive sub-ice shelf system. Oceanic cavities below ice shelf systems are uniquely different from open ocean environments in their dependence on in situ chemosynthesis and on lateral advection of food sources from open-water areas, rather than on vertical fluxes of phytoplankton-derived detrital matter[37]. We

estimate that the waters sampled at the borehole location have been in the cavity for as much as four years prior to sampling; this is up to 10-20-fold longer than the time predicted for marine snow from the ocean surface to reach the abyss (~6000 m[38],). Likewise, the heterotrophic production rates measured in this study and at borehole J9[10] were among the lowest measured in marine ecosystems, including environments with similar temperatures[39]. It has been suggested that production rates are highly influenced by the supply and concentration of labile dissolved organic material[39], and thus the water column beneath the ice shelf is predicted to be highly oligotrophic with respect to labile organic matter.

Based on these heterotrophic rates and assuming a heterotrophic prokaryotic growth efficiency of ~5% (typical of deep oceanic waters, e.g.,[40].), we estimate a total organic carbon demand (i.e., the combined carbon incorporation into biomass and respiration) of ~6–12 $\mu mol$ C $m^{-3}$ $d^{-1}$. This total carbon demand is in the same range as the carbon fixation rates reported from the environment beneath the J9 borehole (8.3 $\mu mol$ C $m^{-3}$ $d^{-1}$[12]). While the contribution of exogenous organic matter remains to be quantified, the close coupling between in situ dark carbon fixation and organic carbon demand suggests that the ecosystem beneath the Ross Ice Shelf is largely sustained by dark carbon fixation. This would differ from deep open ocean environments, where heterotrophic carbon demand significantly relies on the vertical fluxes of particulate organic carbon generated in the euphotic layer[41,42].

Our multi-omic results support this hypothesis, while uncovering the mediators and pathways responsible for the autotrophic and heterotrophic activities under the Ross Ice Shelf (Fig. 6). Among the lineages represented by MAGs and SAGs with the highest transcriptional activity are those originating from the chemolithoautotrophic genera *Nitrosopumilus* and *Nitrospina*. Overall, this agrees with previous reports that aerobic ammonium-oxidizing microorganisms are widespread in Antarctic marine environments (e.g.,[43]) and that ammonium oxidation occurs beneath Antarctic shelves and sea ice[12,44]. These and other inferred facultative chemolithoautotrophs (such as facultative sulfur-oxidizing bacteria) are likely to be responsible for dark carbon fixation rates previously observed beneath borehole J9[12] and thus provide a supply of organic carbon to an ecosystem shielded from sunlight.

The importance of dark carbon fixation has been recognized in various oceanic regions during the polar winter. Microbial lineages (e.g., *Nitrospina*, *Nitrosopumilus*, SAR324, and Marinisomatota[45–47]) and enzymes (such as those mediating ammonium, nitrite, and sulfur oxidation[48]) that mediate chemolithoautotrophy have been observed to increase in Antarctic waters during the transition to the winter season. Likewise, comparable lineages and genes capable of sulfur compound oxidation have been detected in winter open waters and the central basin under the Ross Ice Shelf. Together with mounting evidence that sulfur compound oxidizers sustain carbon fixation in the wide dark open ocean (e.g.,[49]) and the diverse sources of reduced sulfur compounds in marine oxic environments (e.g.,[50]), it is plausible that these clades can also contribute to chemoautotrophy in the oceanic cavity beneath the Ross Ice Shelf.

It is likely that ammonium is a primary energy source sustaining primary production in aphotic Antarctic waters. Consistent with this idea, ammonium oxidation rates have been reported to be higher in Antarctic coastal waters during the austral winter and to significantly support the heterotrophic demand[43]. In the absence of direct rate measurements in this study, we estimated the ammonium oxidation rates potentially supported by the standing ammonium concentrations in the water column. Our estimates for the basal layer (~90 nM $NH_4^+$ $d^{-1}$) are in accordance to rates measured in the Southern Ocean (62 nM $NH_4^+$ $d^{-1}$) with

comparable ammonium concentrations (0.7 μM $NH_4^+$;[43]), and could support the heterotrophic demand in the oceanic cavity under the shelf (Supplementary Notes). These estimates suggests that the microbial communities beneath the Ross Ice Shelf can sustain ammonium oxidation at similar rates to those in the winter Antarctic Ocean and have the potential to be significant primary producers.

The ammonium profile beneath the Ross Ice Shelf is intriguing. Contrary to other nutrient concentrations measured (which do not vary significantly through the water column), ammonium concentrations are significantly higher in the ice basal boundary layer compared to the deeper water samples, but comparable to those in the periphery of the shelf[4]. This profile (exclusive for ammonium with respect to other nitrogen species) is consistent with the reports beneath borehole J9[12]. The proposed circulation model beneath the shelf[13], by which the cavity is filled southward by dense water masses that reach its interior via deep cavity circulation, renders it unlikely that the high ammonium concentrations detected in the fresh, northward flowing waters beneath borehole HW2D or J9 originate from the open Ross Sea.

If externally sourced, nutrient concentrations would be expected to be highest in deeper waters, or else be homogenized in the water column as water masses evolve and mix in the cavity. The latter appears to be the case for the other nutrients measured in this and the J9 expedition. The exception observed in the ammonium profile suggests that this compound could be sourced beneath the ice shelf. In particular, terrestrial-origin sediments in the basal ice layer may be a significant source of ammonium to the seawater circulating beneath. Deployment of cameras at HWD2 revealed sedimentary englacial debris in the lower 20 meters of the ice shelf[13]. While ice melting and freezing can plausibly result in the rainout of the pellets in a sub-ice-shelf cavity, we did not witness this effect; no sediments were retrieved from the pumping samples and the microbial communities sequenced from the englacial debris and the water column were unrelated (Supplementary Fig. 2). However, temperature and salinity data from our study site (Fig. 1b,[13]) clearly showed ice-shelf basal melting and a supply of freshwater to the upper region of the water column, a phenomenon that could result in the observed replenishment of ammonium concentrations in this system. In free-floating sea ice, as well as in subglacial lakes, ammonium enrichments have been traditionally attributed to wet and dry atmospheric deposition, as well as in situ organic matter regeneration in brine channels, especially within older and thicker ice[51–53]. The latter may be also a mechanism for ammonium accumulation in deep layers of the ice shelf[54], subject to solubilization and transport by fresh melt water. If such is the case, the ammonium transported by the ice basal boundary layer could be sourced locally (at borehole HWD2) or elsewhere upstream. Dissolved nutrients in the ice sheet or englacial debris are eventually diluted as they circulate the interior of the shelf[54], which could explain the observed higher concentrations in the water column from borehole J9[12], 330 km upstream from our study site. While the driving factors of the nutrient profile in the water column remain unclear, the tenfold decrease in ammonium concentrations correlate with changes in relative transcriptional activity of the ammonium-oxidizing genus *Nitrosopumilus* (Supplementary Fig. 4). As described in Supplementary Notes, we observed depth-related differences in microbial community composition, metabolic capabilities, and gene expression, though additional depth profiles would be required to confirm this.

The community members with highest relative abundance and transcriptional activity throughout the water column included nitrifying autotrophic taxa and organoheterotrophic bacteria (Supplementary Figs. 4, 5 and 6). It is likely that the genomes with highest relative transcriptional activity represent two opposite

adaptative strategies to the conditions beneath the Ross Ice Shelf. Based on the proportion of their genome expressed, nitrifiers are likely to effectively exploit the surrounding environment by expressing a reduced set of genes encoding a few metabolic pathways. The opposite is observed in the highly expressed heterotrophic clades (Supplementary Fig. 4). By expressing up to 95% of their genome (e.g., in members of Latescibacterota and Verrucomicrobiota), the transcriptional effort of the latter is spread across a variety of process and in particular, to the exploitation of multiple substrates. These observations are consistent with previous studies combining expression and genomic datasets, which suggest that activity levels, substrate utilization and transcriptome diversity may be linked in defining ecological niches of microbial communities[55,56].

In particular, our results suggest that the most active heterotrophic organisms are adapted to degrade complex organic compounds, including most of the enriched phyla in this environment, such as Myxococcota and Planctomycetota. Their capacity to degrade complex organic material from a range of sources, including potentially of both autochthonous and allochthonous origin, likely confers a major selective advantage in this highly oligotrophic ecosystem. Heterotrophy based on the consumption of recalcitrant dissolved organic carbon has been considered as one possibility for sustaining the oceanic Antarctic winter food web[57], and could also be an additional support for life under the Ross Ice Shelf. Unlike organic carbon in Antarctic winter waters, which may have accumulated during the highly productive summer season, organic substrates beneath the Ross Ice Shelf potentially consist of vertically transported exudates and necromass derived from lithoautotrophic primary producers, but also recalcitrant complex organic compounds laterally transported from the Ross Sea into the shelf cavity. Decomposition of phytoplankton entering the shelf cavity is estimated at a scale of ~10 years[4]. Together with previous reports of diatoms in below-shelf waters[9], this indicates that some photoautotrophically-derived organic matter can reach the center of the oceanic cavity. However, the metagenomes suggest that photosynthetic eukaryotes (i.e., class Bacillariophyceae) make a small fraction of the eukaryotic community (0.05 %); this finding is also consistent with undetectable concentrations of chlorophyll a beneath borehole J9[12]. Despite potentially serving as a substrate for organoheterotrophs beneath the ice shelf, phytoplankton are therefore unlikely contributors to the dissolved organic matter pool, whereas detrital sources of bacterial substrates may be more important. Further work is now needed to discriminate organic matter sources and nutrient exchange processes within the shelf.

Overall, microorganisms under Antarctica's ice shelves can thrive in some of the coldest and possibly carbon-limited marine waters, while playing a crucial role in the remineralization of nutrients to the Southern Ocean. Our results not only suggest that the waters below the Ross Ice Shelf are driven by chemolithoautotrophic processes, but also uncover the mechanisms responsible for sustaining that activity[58]. Alongside other recent reports of oceanic dark carbon fixation,[27,49,59], this study also emphasizes the importance of inorganic energy sources in driving marine communities in the absence of photosynthesis. Finally, our results suggest that ammonium associated with fresh melt waters at the base of the ice is an important supply of inorganic electron donors supporting chemolithoautotrophy, and thus has a significant influence in the composition and activity of the microbial community. Ocean-driven basal melting, a source of freshwater and thus potentially of ammonium in the sub-ice cavity, may increase in a warming climate scenario[60]. Assuming that our observations are representative of the central region of the cavity under the Ross Ice Shelf, increased basal ice melting could result in an increased vulnerability of communities supported by sub-ice shelf processes[61], potentially leading to shifts in

the relative biogeochemical importance of chemolithoautotrophic processes in this extensive ecosystem. These insights emphasize the importance of baseline data from existing sub-ice shelf ecosystems, such as the Ross Ice Shelf, to inform the prediction of biogeochemical impacts of climate change in the Southern Ocean.

## Methods

**Site selection and description**. Sampling took place in December 2017 and was conducted by members of the Aotearoa New Zealand Ross Ice Shelf Program. Samples were collected from the sub-shelf water column at a site in the central region of the ice shelf, borehole HWD-2 (Latitude -80.6577 N, Longitude 174.4626 W), ~300 km from the shelf front and 330 km northwest of borehole J9 (Fig. 1a). The sampling site is near the glaciological boundary between ice originating from the West Antarctic Ice Sheet and ice flowing from East Antarctica through Transantarctic Mountain glaciers (Fig. 1a). Sediment of terrestrial origin was observed in the lowermost ~60 m of the ice.

**Hot water drilling and sampling**. A hot water drilling system built and operated by the Victoria University of Wellington Drilling Office was used to bore through the ice shelf, creating an access borehole with a maximum diameter of 30 cm. The borehole was used for direct sampling of water and sea floor sediments, and to conduct in situ measurements in the water column. These activities were conducted inside a custom-built tent that facilitated 24-h operations in any weather conditions. Seawater samples were obtained from three depths (400 m, 550 m, and 700 m from the top of the shelf, which correspond to 30 m, 180 m, and 330 m deep from the bottom of the ice shelf, respectively). These were chosen to characterize the water column under the Ross Ice Shelf while keeping the sampler ca. 40–50 m away from the seafloor and from ice crystals and sediment in the ice-shelf basal layer. The drilling water was fresh (<15 psu) and relatively warm (between −1 and +1 °C), so it remained stably floating in the borehole and did not sink into deeper layers. This, together with the advection of seawater below the ice shelf, precluded any contamination of collected seawater with the drilling water (Supplementary Fig. 2a, b). The lack of intrusion of the freshwater used for the drilling was routinely checked by salinity and temperature-depth profiles.

Samples were collected by in situ filtration using a McLane WTS-LV-Bore Hole filter pump fitted with a 142 mm diameter, 0.22 μm pore-size filter (Supor membrane filters, Pall Corporation). Before and after deployment, the filter holder was thoroughly cleaned to avoid sample cross-contamination. The pump head interior was also flushed after every deployment with fresh water to prevent salt crystal formation and sample contamination. This sampling approach was aimed at obtaining the most realistic representation of the microbial community's composition and activity with the minimum possible sampling biases. Approximately 200 L of water were filtered at each depth within ca. 2 h. Thereafter, filters were placed in sterile Petri dishes and divided into seven sections using sterile scalpels and transferred to cryovials. The filtered, frozen samples were directly stored in zip lock bags in a 3 m deep borehole drilled into the cold surface snow layer until transported to Scott Base (and further airplane transport to New Zealand). The temperature of the samples deposited in the storage borehole remained stable ranging mostly between −27 °C and −28 °C (Supplementary Fig. 17). These samples were used for 16S rRNA amplicon sequencing, metagenomics, and metatranscriptomics.

Water samples (150–300 mL) were also collected at the same three depths using the McLane WTS-LV-Bore Hole pump without a filter-holder in order to further minimize sample contamination. Once the pump was brought up, it was run in reverse to collect the water, but excluding the first 30–60 mL of water (used for rinsing). Water samples for inorganic nutrient analyses were filtered through combusted Whatman GF/F filters, collected in acid-cleaned HDPE bottles, and stored frozen until analysis in the home laboratory, following procedures recommended by the Joint Global Ocean Flux Study (JGOFS[62]). The liquid samples for the determination of microbial cell abundance, prokaryotic heterotrophic production, and the generation of single-cell amplified genomes (SAGs) were collected in acid-cleaned Nalgene™ opaque amber HDPE bottles, stored at 2 °C, and transported within 48 h to Scott Base to perform further laboratory analyses. The samples were imported to New Zealand under Ministry for Primary Industry permit number 2017063583 (Permit to import Restricted Biological Products of Animal Origin) issued to the University of Otago Department of Marine Science.

To check for potential contamination, samples were also collected from the following sites: freshly melted snow nearby the camp area, drilling water from a reservoir tank, and sediments dislodged from the ice shelf (identified as englacial debris) and collected with the reaming tool. Water samples were filtered onto 0.22 μm polycarbonate filters (47 mm filter diameter, Millipore), and all samples were stored in cryovials and frozen.

**Physicochemical measurements**. A SBE 19plusV2 SeaCAT Profiler CTD (Seabird Electronics, Inc.) was used to measure temperature, salinity and depth within the borehole and in the water under the Ross Ice Shelf for a detailed characterization of the water column. Furthermore, a self-contained single channel logger (RBR Solo) was attached to the frame of the WTS-LV-Bore Hole pump (at the opposite side of

the water intake) for an accurate determination of the temperature and depth of the sampling casts. Samples for determining the concentrations of nitrate, dissolved reactive phosphorus (phosphate), ammonium and $SiO_2$[62] were colorimetrically analyzed using flow-injection analysis on a Lachat Auto-analyzer according to methods described elsewhere[63]. Measurements of nutrient concentrations were routinely corrected with reference blank solutions in each sample run. No anomalies were detected in the blanks, indicating no source of detectable contamination during the measurements.

**Prokaryotic abundances and heterotrophic production**. Prokaryotic abundance was determined by flow cytometry. Samples (1.6 mL) were preserved with glutaraldehyde (2% final concentration), left at 4 °C in the dark for 15 min, flash-frozen in liquid nitrogen, and stored at −80 °C until analysis. Prior to analysis, the fixed samples were thawed, stained in the dark with a DMS-diluted SYTO-13 dye (Molecular Probes Inc., 2.5 μM final concentration) for 5 min, and run on a BD AccuriTM flow cytometer with a laser emitting at 488 nm wavelength. Samples were run at low or medium speed until 10,000 events were captured. A suspension of yellow–green 1 μm latex beads ($10^5$–$10^6$ beads mL$^{−1}$) was added as an internal standard (Polysciences, Inc.).

Prokaryotic heterotrophic activity was estimated via the incorporation of $^3$H-leucine using the centrifugation method[64]. $^3$H-leucine (Perkin-Elmer, specific activity 169 Ci mmol$^{−1}$) was added at saturating concentration (40 nmol L$^{−1}$) to triplicate 1.2 mL subsamples. Controls were established by adding 120 μL of 50% trichloroacetic acid (TCA) to triplicate control tubes 10 min prior to radioisotope addition. The microcentrifuge tubes were incubated in the dark at 4 °C for 48 h. Incorporation of leucine in the quadruplicate tubes per sample was terminated by adding 120 μL ice-cold 50% TCA. Subsequently, the samples and the controls were kept at –20 °C until centrifugation (at ca. 12,000 × g) for 20 min followed by aspiration of the water. Finally, 1 mL of scintillation cocktail was added to the microcentrifuge tubes before determining the incorporated radioactivity after 24–48 h on a Tri-Carb 2000® Liquid Scintillation Counters scintillation counter (Perkin-Elmer) with quenching correction. The blank-corrected leucine incorporation rates were converted into prokaryotic heterotrophic production (PHP) using the theoretical conversion of 1.55 kg mol$^{−1}$ leucine incorporated[65–67]. The rates of leucine incorporation obtained at the incubation temperature (4 °C) were converted to the in situ temperature of -2 °C using an activation energy of 72 kJ mol$^{−1}$[67].

**Single cell genomics**. Sample collection and analyses were performed as described previously[27], see Supplementary Methods for full description. Briefly, triplicate seawater samples (1 mL) were transferred to a sterile cryovial containing 100 μL of glyTE (20 mL of 100 × TE buffer pH 8.0, 60 mL Milli-Q water and 100 mL of molecular-grade glycerol), and samples were stored at –80 °C until analysis. SAG generation was performed at the Single Cell Genomic Center at Bigelow Laboratory for Ocean Sciences (SCGC) using fluorescence-activated cell sorting and WGA-X genomic DNA amplification. Paired-end Illumina libraries were created with Nextera XT (Illumina), sequenced with NextSeq 500 (Illumina) and de novo assembled using a workflow based on SPAdes[68] as previously described[69]. The quality of the sequencing reads was assessed using FastQC v0.11.7 (https://www.bioinformatics.babraham.ac.uk/projects/fastqc/) and the quality of the assembled genomes was determined using CheckM v.1.0.7[70] and tetramer frequency analysis[71]. This workflow was evaluated for assembly errors using three bacterial benchmark cultures with diverse genome complexity and %GC, indicating no non-target and undefined bases in the assemblies and average frequencies of mis-assemblies, indels and mismatches per 100 kbp: 1.5, 3.0 and 5.0[69]. Functional annotation was first performed using Prokka[72] with default Swiss-Prot databases supplied by the software. Prokka was run a second time with a custom protein annotation database built from compiling Swiss-Prot[73] entries for Archaea and Bacteria.

**DNA extraction, 16S rRNA gene amplicon and metagenomic sequencing**. DNA was extracted using a PowerSoil® DNA Isolation Kit (MoBio, Carlsbad, CA, USA). The manufacturer's protocol was modified to use a Geno/Grinder for 2 × 15 s instead of vortexing for 10 min and a final elution of 50 μL solution C6 (sterile elution buffer, 10 mM Tris) was used. DNA concentration was measured using a Nanodrop spectrophotometer (Thermo Fisher). The median 260/280 nm wavelength ratio was 1.5 with a lower quartile of 1.4 and an upper quartile of 1.7. Extractions were performed in triplicate for each depth under the Ross Ice Shelf (total of 9 samples) for subsequent amplicon and metagenomic sequencing.

16S rRNA gene amplicon sequencing was carried out using the Earth Microbiome Project[74] protocols and standards (http://earthmicrobiome.org/protocols-and-standards/16s/), which include the following modifications to the original 515F–806 R primer pair[75] (the updated sequences, 5′- 3′, are as follows: 515 F: GTGYCAGCMGCCGCGGTAA; 806 R: GGACTACNVGGGTWTCTAAT). In brief, degeneracy was added to both the forward and reverse primers to remove known biases against Crenarchaeota/Thaumarchaeota (515 F, also called 515F-Y[76]) and the marine and freshwater Alphaproteobacterial clade SAR11 (806 R[77],). All amplicons (independent replicates) were run on an Illumina (Foster City, CA, USA) MiSeq 250 bp × 2 run. For metagenomic sequencing, Thruplex DNA libraries

(~300 bp inserts) were created from each individual DNA extraction and sequenced in an Illumina HiSeq 2500 platform (2 × 125 bp).

**RNA extraction and metatranscriptomic sequencing.** RNA was extracted following the RNeasy mini kit (Qiagen, Hilden, Germany) procedure and the ethanol precipitation protocol. The remaining DNA was removed with TurboDNase (Invitrogen, Carlsbad, CA, USA) and the efficiency of removal was tested with PCR. Enrichment of RNA was performed with 20 μL of sample RNA following the procedures of the MICROBEnrich (Ambion, Austin, TX, USA) and MICROBExpress (Ambion, Austin, TX, USA) kits. Thereafter, the MessageAmp II-Bacteria kit (Invitrogen) was used to improve the subsequent amplification and purification: enriched RNA was reverse transcribed to cDNA, which was in vitro transcribed back to amplified RNA (aRNA) using the mentioned kit. Quantifications were simultaneously run with a Nanodrop spectrophotometer (Thermo Fisher) and a Qubit fluorometer (Invitrogen, Carlsbad, CA, USA) using the RNA HS Assay kit and an RNA profile generated with a Bioanalyzer 2100 (Agilent Technologies, Böblingen, Germany). aRNA was shotgun sequenced directly in an Illumina HiSeq4000 platform (CNAG, Barcelona, Spain), generating between 28–35 Gb of 2 × 101 bp reads per sample.

**16S rRNA gene amplicon profiling.** Paired-end 16S rRNA gene amplicon sequences were processed on the QIIME2 platform using the DADA2 pipeline to resolve exact amplicon sequence variants[78,79]. Raw reads were demultiplexed, yielding 302,585 reads across 16 samples. Quality plots were generated and sequences failing to pass an average base call accuracy of 99% (Phred score 20) were excluded. Low quality regions of each sequence were removed by trimming the first 13 bases of the forward and reverse reads and truncating at 150 base pairs before de-noising with DADA2 using the function *qiime dada2 denoise-paired* with default parameters. The final dataset contained 1228 amplicon sequence variants (ASVs) with a total frequency of 271,736. Taxonomic assignment was performed by using a Genome Taxonomy Database classifier built for the QIIME2 platform, using the SSU sequence files from GTDB ssu_r86.1_20180911 (https://osf.io/25djp/wiki/home/). The classifier was first spliced to the 515 F/806 R primer pair using the *qiime feature-classifier extract-reads*, and trained using the *qiime feature-classifier fit-classifier-naive-bayes* command in QIIME2[79]. The trained classifier was then used to assign the taxonomy to the ASV features using our representative reads via the function *feature-classifier classify-sklearn*. No sequence overlap was observed between below-shelf waters with those of control samples (e.g., drilling fluid, sediment recovered from basal ice on the shelf, snow at the camp site) (Supplementary Fig. 2), confirming absence of contamination in the water column samples.

**Metagenomic community profiling.** Raw metagenomic and metatranscriptomic paired-end reads were quality-assessed with FastQC v0.11.7 and MultiQC v1.0[80]. BBDuk v38.51 from the BBTools suite (https://sourceforge.net/projects/bbmap/) was used to trim adapter sequences, remove reads corresponding to Illumina's PhiX sequencing control, trim low-quality bases (minimum quality score 20), and discard short sequences (minimum length 50 bp). The metatranscriptome reads were further processed with SortMeRNA v2.1b[81] to remove reads corresponding to prokaryotic and eukaryotic ribosomal RNA, followed by BBDuk to filter low-complexity reads (entropy threshold 0.05).

In addition, taxonomic profiling of bacteria, archaeal, and eukaryotic communities was performed with 16S rRNA gene sequences extracted from metagenomic reads (miTags) using a previously described protocol[19]. miTags were also extracted from bathypelagic samples from the Malaspina Circumnavigation expedition[23], metagenomic surveys in the Arctic and Southern Ocean[21], as well as metagenomic datasets from polar regions obtained from the TARA Ocean Expedition[22]. This allowed comparing these datasets to available miTags from epipelagic and mesopelagic samples from the TARA Ocean Expedition[20]. Extracted 16S and 18S rRNA gene reads were mapped to the SILVA non-redundant SSU Ref database (v.138)[82] and assigned to an approximate taxonomic affiliation (nearest taxonomic unit, NTU) using PhyloFlash v3.0[83] (http://github.com/HRGV/phyloFlash).

Bacteriophage prediction was based on identifying viral signals in the metagenomic-assembled contigs (described below) using VirSorter[84]. In brief, viral-like genes were identified against a curated virome database[84] and a set of single-amplified viral genomes[85]. Abundance of viral contigs was estimated by recruitment of metagenomic reads to viral contigs and calculation of contig coverage. Open reading frames (ORFs) were detected and translated with Prodigal v.2.6.3[86]. Taxonomic classification of the translated sequences was based on sequence homology search[87] against the Uniref 100 viral database (http://virome.dbi.udel.edu; e-value < $10^{-5}$) and used to obtain taxonomy classification of viral contigs with the *anvi-import-taxonomy* function from Anvi'o v.5.2[88]. The metagenomic reads were mapped to the obtained viral contigs using Bowtie 2[89] (local alignment, sensitive setting). Coverage of viral contigs was calculated by metagenomic read recruitment using Anvi'o.

**Alpha- and beta-diversity analyses of 16S rRNA amplicons and extracted miTAGs.** All statistical analyses were carried out in R v3.5.3. Data manipulation was performed using the R package tidyverse and all visualizations were made

using ggplot2. Community richness and beta-diversity was calculated using the R packages Phyloseq[90] and Vegan v2.5-6[91]. In total, nine samples representing a triplicate of depth profiles were used for downstream diversity analysis of ASVs (Supplementary Fig. 3, Supplementary Data 3). Rarefaction curves were constructed to confirm that sequencing depth adequately captured richness in each sample and rarefied using the Phyloseq *rarefy_even_depth* function with a sample size of 15,400, which represented the minimum sequencing depth to retain 100% of samples used for downstream analysis. Observed richness (counts) and estimated richness (Chao1) was calculated using the estimate richness function in Phyloseq. Normality of the distribution of alpha-diversity estimates was confirmed using a Shapiro-Wilk test and a one-way analysis of variance (ANOVA) to test for significant differences in richness across depth profiles. As a post-hoc, a Tukey multiple comparison of means was used to confirm which pairs of sites showed significant differences. For beta-diversity analysis on amplicon and miTag data, Bray Curtis distance matrices were calculated in Vegan and visualized using a principal coordinate analysis (PcoA). Independent permutational analysis of variance (PERMANOVA) based on the Bray-Curtis dissimilarities values were calculated with the *adonis* function in Vegan (999 random permutations), to test for significant differences in community structure between depth profiles. Finally, a beta-dispersion test (PERMDISP) was applied to confirm that observed differences were not influenced due to dispersion. As a post-hoc evaluation of taxa responsible for differences in microbial community structure, we performed an indicator species analysis. We used the indicator value method[92] to calculate indicator values using the R package indicspecies. An individual ASV was considered a valid indicator species if the p value was < 0.05 and the Test statistic (the indicator value) was 0.5 or greater, based on 1000 random permutations[93]. IndVals were compared between two groups, basal layer (30 m) and mid-column samples (180 m and 330 m), with the *multipatt* function in the R Indicspecies package (with the option control = how(nperm = 999)). This function uses an extension of the original Indicator Value method: it looks for indicator species of both individual site groups and combinations of site groups[94].

Counts per NTU (at species-level resolution) of extracted miTAGs were used for comparative analyses between communities under the Ross Ice Shelf and other oceanic samples. Only bacterial and archaeal species with >4 reads per sample were included in the analyses. Samples were divided into four groups, according to sampling depth or location: below-shelf ocean cavity (depth 30–330 m, $n = 9$), epipelagic (depth <200 m, $n = 169$), mesopelagic (depth ~200–1000 m, $n = 60$), and bathypelagic (depth 1000–4000 m, $n = 54$). The Vegan function vegdist was used to calculate a Bray-Curtis dissimilarity matrix between all samples, which was visualized by hierarchical cluster analysis (average linkage method, function hclust in Vegan). Significant differences ($p < 0.05$) between relative abundances of taxa from deep (>200 m) open ocean communities worldwide and below-shelf communities were confirmed using a non-parametric one-way analysis of variance (Kruskal-Wallis test, function *kruskal.test()* in R base).

The following comparisons were restricted to two groups from deep, polar environments: samples from mesopelagic and bathypelagic polar environments ($n = 42$) and samples from the below-shelf cavity ($n = 9$). As distance-based multivariate methods can confound the within- and between-group effect size and fail to account for the mean variance relationship[95], a generalized linear model (GLM) approach was used via the R package mvabund[96]. A multivariate model was fitted using the *manyglm* function and negative binominal distribution. To test the multivariate hypothesis of whether species composition varied across sub-ice and open water, the *anova* function was used which performed an analysis of deviance using likelihood ratio tests (LRT) and PIT-trap resampling of p values using 1000 iterations. To further examine which taxa contribute to compositional changes, a series of univariate tests were performed on each taxon using the *p.uni = "adjusted"* argument in the anova function. IndVal values were also calculated, using the same parameters described above, to identify which species contributed most to the differences between sub-ice environments and deep open ocean waters, Further, an additional post hoc test for between-group differences was performed with analysis of similarity percentages (simper[97],) on a Bray-Curtis dissimilarity matrix calculated as described above.

**Metagenomic assembly and binning.** For assembly, metagenome paired-end reads were error corrected using Bayes Hammer implemented in SPAdes v.3.0.0[68], merged with BBmerge v.36.32[98] and normalized to a kmer depth of 42 with BBnorm v.36.32, from the BBtools program suite. Co-assembly of metagenomes was performed with MEGAHIT v.1.1.1[99] with merged and unmerged reads. Metagenomic reads were mapped back to the co-assembly (min. length 1 kb) using BBmap v.36.32[100] to calculate differential coverage across all samples.

Contigs were binned with MetaWatt v.3.5.3[101], MaxBin v.2.2.7[102] and MetaBAT v.2.12.1[103]. Bins were automatically de-replicated and aggregated with DasTool[104], then manually inspected and refined with Anvi'o v.5.2[88]. Bins classified as Archaea, Gammaproteobacteria, Deltaproteobacteria, Gemmatimonadota, Actinobacteriota, and Chloroflexota were selected from the bulk co-assembly and used for read recruitment with a minimum identity of 70% using BBmap v.36.32. This led to less complex subsets of reads for subsequent re-assembly with a more thorough assembler (SPAdes). For each taxonomic group a separate re-assembly with SPAdes v.3.0.0 was performed followed by a new round of binning as described above and manual refinement in Anvi'o. This procedure

improves assembly (i.e., number of scaffolds reduced) and consequently bin metrics such as contig length and purity of bins[105]. Completeness and quality of final assemblies were assessed by CheckM v.1.0.7[70], with bins with >50% completeness and <5% contamination (i.e., high and medium quality bins) retained for further analysis[25].

**Genome de-replication, classification, and phylogenetic analysis.** Metagenomic bins and single-cell-assembled genomes with >50% completeness were defined as MAGs and SAGs, respectively, and collectively as 'genomes' for simplicity. Comparison and de-replication of genomes were performed with dRep pipeline[106]. In brief, genomes were grouped at an average nucleotide identity (ANI) of 99%. Representative genomes from each cluster were selected based on the highest 'genome score'[106]. This analysis provided a de-replicated genomic database of population genomes. BBmap and samtools were used to recruit reads from the metagenomes (97% identity), and Anvi'o was used to calculate the interquartile (Q2Q3) mean coverage of the de-replicated genomes across samples. On average, 50–60% of each sample's metagenomic reads mapped to the metagenomic and SAG contigs.

MAGs and SAGs were taxonomically assigned using the tool GTDBTk v.0.0.6 (release 80, www.github.com/Ecogenomics/GtdbTk) in accordance to the Genome Taxonomy Database[107] (Supplementary Data 4). Phylogenetic tree construction for all 235 MAGS/SAGS was performed using ribosomal protein sequences retrieved from CheckM v.1.0.7[70] (Fig. 3). The concatenated marker sequence for each genome was aligned using MAFFT[108] and an approximate maximum-likelihood phylogenetic tree was generated using FastTree 2[109] with default parameters. The tree was then visualized and annotated using the web-based tool iTOL v.6 (https://itol.embl.de).

**Metabolic profiling of MAGs, SAGs, and assembled unbinned reads.** ORFs in binned and unbinned contigs were predicted using Prodigal v.2.6.3.[86], with default noise-cut-offs followed by manual filtering using HMM cut-off scores previously described[110]. The predicted ORFs were automatically annotated with the standard RAST annotation pipeline[111], and against the Pfam (release 32.0)[112] and TIGRfam (release 15.0)[113] HMM models using Interproscan 5[114].

Phylogenetic trees were constructed to validate findings and to determine which protein classes / lineages were present in the Ross Ice Shelf (Supplementary Figs. 7–16). Trees were constructed for AmoA, NxrA, HbsT, RbcL, AclB, DsrA, Sqr, SoxB, CoxL, and the group 1 h [NiFe]-hydrogenase (HhyL). In all cases, protein sequences retrieved from the MAGs, SAGs, and metagenomic assembled reads by homology-based searches were aligned against a subset of reference sequences from a custom database containing 51 proteins (available at https://doi.org/10.26180/c.5230745) using ClustalW in MEGA7[115]. Evolutionary relationships were visualized by constructing maximum-likelihood phylogenetic trees. Specifically, initial trees for the heuristic search were obtained automatically by applying Neighbor-Join and BioNJ algorithms to a matrix of pairwise distances estimated using a JTT model, and then selecting the topology with superior log likelihood value. All residues were used, and trees were bootstrapped with 50 replicates.

Annotation of carbohydrate active enzymes (CAZymes) was performed by protein search against the CAZyme HMM database (dbCAN HMMdb release 8.0) following the dbCAN2 CAZyme annotation pipeline[116], with stringent parameters for all CAZyme classes (E-value <1e$^{-15}$ and coverage >0.35). We quantified the number of genes in each genome encoding for different glycosyl hydrolases (GH), glycosyl transferases (GT) and containing carbohydrate binding domains (CBD) (Supplementary Data 7). Heatmaps for the 50 genomes with highest GH diversity were generated in R with ggplot2 (Fig. 6), representing their abundance in the metagenome and the metatranscriptome (as described in the section below).

**Comparison of abundance and expression of assembled reads.** To analyze the expression of annotated ORFs, pre-processed metatranscriptomic paired reads were merged with BBmerge[98]. Merged and unmerged non-rRNA sequences were mapped to the metagenomic and SAG contigs (99% id) with BBmap (on average, 60% of each sample's reads were successfully assigned). Quantification of mapped reads per identified gene was performed with the function *featureCounts* of the R Subread package[117]. The transcript abundance of each ORF was converted to transcript per million (TPM) (Eq. (1)) for each sampled depth.

$$TPM = A * 1/\Sigma A * 10^6 \quad (1)$$

where $A$ = reads mapped to gene/gene length (kbp).

To minimize systematic variability of individual gene abundance, the genome interquartile (Q2Q3) mean coverage (or, for unbinned contigs, the contig's coverage) was used to define gene abundance in the metagenome. Gene coverage was then converted to counts per million (CPM), to allow for direct comparison with TPM.

$$CPM = B * 1/\Sigma B * 10^6 \quad (2)$$

where $B$ = gene coverage.

Data from sample replicates were combined for the above calculations.

**Metabolic profiling of unassembled metagenome and metatranscriptome reads.** The abundance of particular metabolic functions independent of assembly was calculated as previously described[118]. Briefly, pre-processed metagenomic and metatranscriptomic reads were aligned using DIAMOND v0.9.24 to the 1 manually curated protein databases described above and to the predicted ORFs that matched the additional 10 HMMs described above (Supplementary Data 6). DIAMOND mapping was performed with a query coverage threshold of >80% and a gene specific threshold of 40% (RHO), 60% (AtpA, AmoA, MmoA, CoxL, NxrA, NuoF and RbcL), 75% (HbsT), 70% (PsbA, YgfK, ARO, IsoA), (80%) PsaA, or 50% (all other databases), with data further parsed to retain only group 1 and 2 [NiFe]-hydrogenase hits. For the metagenomic data, forward reads with at least 124 bp in length were used. For the metatranscriptomic data, paired-end reads were merged with BBMerge v38.51 and merged reads of at least 124 bp in length were used. Data from sample replicates were combined for this analysis. The abundance of each gene was converted to reads per kilobase million (RPKM).

$$RPKM = X/\text{total sample reads} * 10^6 \quad (3)$$

where $X$ = reads aligned to a gene/ gene length (kbp).

The gene abundances in RPKM from the metagenomic data were further used to estimate the proportion of the community encoding these functions. The processed metagenomic reads were aligned to each of the 14 universal single-copy ribosomal marker genes available in SingleM (https://github.com/wwood/singlem) with DIAMOND using a query coverage threshold of 80%. Alignments with a bitscore below 40 were removed; the alignment counts were converted to RPKM as described above and averaged across the 14 genes to represent the abundance of a universal single-copy gene. Metabolic gene RPKM values were divided by this value to obtain the average gene copies per organism in each sample (abundance relative to a single-copy gene). Heatmaps representing the community percentage (metagenomic data) and RPKM abundance (metatranscriptomic data) were generated in R with ggplot2 (Fig. 4b). Where genes within the same pathway are collapsed together, the values (community percentage or RPKM) are summed.

**Reporting summary**. Further information on research design is available in the Nature Research Reporting Summary linked to this article.

## Data availability

The data and code underlying Fig. 2a, c, d are provided in the github repository https://github.com/ClaMtnez/Ocean_tags. The data underlying Figs. 3, 4 & 5 and Supplementary Figs. 1, 4 & 5 are provided as a Supplementary Data Files. The sequence data generated in this study have been deposited in the EMBL Nucleotide Sequence Database (ENA) database under Bioproject PRJEB35712 (metagenomic and metatranscriptomic raw reads, metagenomic and metatranscriptomic assemblies, metagenomic assembled genomes, and single-cell amplified genomes) and in the NCBI Sequence Read Archive (SRA) under Bioproject PRJNA593264 (16S rRNA gene amplicon reads). The following public databases were used in this study: Swiss-Prot database, https://www.uniprot.org/, release-2018_10; Genome Taxonomy Database, https://gtdb.ecogenomic.org/, release 80; SILVA non-redundant SSU Ref database, https://www.arb-silva.de/, v.138; UniRef 100 VIROME database, http://virome.dbi.udel.edu; Greening lab metabolic marker gene database, https://doi.org/10.26180/c.5230745; CAZyme HMM database, https://bcb.unl.edu/dbCAN2/, v.8.0; Pfam HMM database, http://pfam.xfam.org/, release 32.0; and TIGRFAM HMM database, https://www.ncbi.nlm.nih.gov/genome/annotation_prok/tigrfams/, release 15.0

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

## Acknowledgements

We are grateful to D. V. Meier and P. M. Leung for insightful discussions. We thank the Victoria University of Wellington Hot Water Drilling Team led by A. Pyne and D. Mendeno for fieldwork support. Staff of the Bigelow Laboratory Single Cell Genomics Center are acknowledged for generating SAG data. We also thank L. Montiel and V. Balagué from the Institut de Ciències del Mar (ICM, CSIC) for extracting RNA, and the CNAG staff for RNAseq library preparation and sequencing. Bioinformatics analyses were performed at the LiSC Cluster (University of Vienna), MARBITS platform (ICM, CSIC) and the MonARCH HPC Cluster (Monash University). This research was facilitated by the New Zealand Antarctic Research Institute (NZARI) funded Aotearoa New Zealand Ross Ice Shelf Programme, the New Zealand Antarctic Science Platform ANTA1801, the Austrian science fond (FWF) project AP3430411/21 (FB) and a Rutherford Discovery Fellowship from the Royal Society of New Zealand (FB), the US National Science Foundation grants DEB-1441717 (RS) and OCE 1335810 (RS), the Simons Foundation Grant 827839 (RS), the Austrian Science Fund project P28781-B21 (GJH), the Spanish Ministry of Science and Innovation (Spanish State Research Agency, https://doi.org/10.13039/501100011033) fellowship RYC-2013-12554 (RL) and projects CTM2015-69936-P (RL) and PID2019-110011RB-C32 (JMG), the NHMRC EL2 Fellowship APP1178715 (CG) and Discovery Project grant DP180101762 (CG), the ARC SRIEAS Grant SR200100005 Securing Antarctica's Environmental Future (SKB), and the H2020 MSCA Individual Fellowship 886198 (CMP).

## Author contributions

F.B., C.H., S.E.M., and C.O. designed field experiments. F.B., S.E.M., C.H., C.O., C.S., and B.T. performed field sampling and measurements. S.E.M. and R.L. performed nucleic acid extraction and library preparation for metagenomics and metatranscriptomics, respectively. R.S. provided single-cell amplified genome sequencing. C.M.P., Z.Z., R.J.L.,

S.K.B,. D.D.C., B.T., J.M.G., F.B., and C.G. analyzed the data. C.M.P., C.G., and F.B. wrote the manuscript with assistance from all coauthors.

## Competing interests

The authors declare no competing interests.
