## [Peer Review File · Nature Communications]

Phylogenetically and functionally diverse microorganisms reside under the Ross Ice ShelfREVIEWER COMMENTS

Reviewer #1 (Remarks to the Author):

In this manuscript the authors conduct the first in-depth characterization of microbial community structure and function below the Ross Ice Shelf. Several techniques were applied including metagenomics, single-celled genomics, metaproteomics, metatranscriptomics, and amplicon sequencing. They identified a community below the RIS that is distinct from other polar seawater communities and deep ocean samples. This work represents an important contribution to our understanding of microbial diversity and biogeochemistry beneath Antarctic ice shelves, an expansive yet understudied environment.

General comments:

*The statistics are presented in a somewhat haphazard manner and insufficiently described in the methods. For example, the p-values are often presented as ($p < 0.05$) instead of exact values, and the indicator species analysis is not described in the methods.

*Were p-values corrected for multiple comparisons?

*Because the proteomic and metatranscriptomic data aren't really being compared to anything (i.e. there is no control) these measures are of minimal value. They don't seem central to the manuscript, and the proteomics data is hardly mentioned. I recommend focusing this manuscript on the amplicons, MAGs, SAGs, which is certainly enough data and quite novel given the environment.

*The best external points of comparison are likely to be Antarctic winter samples. I believe all the cited studies are from summer. I'm not aware of any modern metagenome studies for winter, but there are some amplicon studies that include winter or early spring samples for the Antarctic (including from the Ross Sea) and these should be brought in.

*No data is available yet at either of the cited BioProjects. Please be sure to include your data so that this can be verified prior to publication. Also, recommend pointing the reader to NCBI BioProject database in case they don't recognize the origin of those project numbers.

Specific comments:

*Line 163 - here and elsewhere, be cautious about characterizing something as "abundant" when you mean relative abundance. Of course since you have abundance data for RIS and open water I encourage you to make the conversion to absolute abundance where appropriate to do so.

Line 206 - Is this remarkable? Isn't gene expression typically dominated by a small but variable fraction of the genome? If anything 25 % seems rather high.

Line 255 - Proteins don't get highly expressed, genes do. Why would cold-shock genes be expressed in such a homogenous environment?

Line 311 - Given the metabolic plasticity of dinoflagellates it seems odd to invoke photosynthesis here, seems far more likely that they are living heterotrophically, regardless of whether endemic or transient.

Line 327 - Remove "of" near end of this line.

Line 411 - This estimate from activation energy requires some justification/explanation.

Line 440 - Those primers have been superseded by others that are more sensitive to SAR11 and archaea (both important for this study) (e.g., Walters et al. 2015, mSystems). Please double check that Caporaso et al. 2012 was in fact the set used, and justify/provide caveats if so.

Reviewer #2 (Remarks to the Author):

The current manuscript by Martínez-Pérez et al. uses combined omics techniques to study the microbial diversity present under Ross Ice Shelf (RIS). I would like to state that I appreciate the difficulties associated with acquiring and analyses of samples originating from such remote places. I also find the topic potentially interesting; however, the manuscript has many drawbacks that I do not believe can easily be remedied. Unfortunately, the authors do not provide convincing (or bluntly any) evidence that substantiates their supposition that ammonia oxidation and nitrite oxidation derived carbon fixation fuels the dark microbial food web. This supposition is indeed the main thread throughout the manuscript; however, there are several side lines that pop up throughout the manuscript without much of an explanation and connection to the rest of the

manuscript, for example consumption of sulfur compounds, recycling of complex organic material or ocean warming. As such, these side lines do not make the manuscript stronger, but very confusing to go through as if the authors could not agree upon what to include in the manuscript. Comments:

- 1) The authors use only three sampling points in their nutrient depth profile. This is incredibly limited and cannot be used to pinpoint any microbial activity to any zone in the depth profile. In order to achieve this a much higher resolution nutrient profile would need to be determined.
- 2) The authors suggest that the highest dark carbon fixation rates are at the sea-ice boundary layer. They infer this based on the high abundance of ammonia-oxidizing microorganisms in this sampling point and that the highest measured ammonium concentrations were here. These measurements and experiments cannot be used to determine ammonia and nitrite oxidation rates or carbon fixation rates. In order to bring the authors' assertions to the level of hard evidence-based facts, ammonia oxidation rates and carbon fixation rates must be measured. The authors should keep in mind that per cell activity is not a constant thing in a culture, let alone in the environment. Furthermore, it depends on many environmental factors ranging from availability of resources to the presence of predators. Therefore, for studies such as these it is paramount to provide direct measurements of the targeted activities rather than just proxies for these activities.
- 3) I am not convinced that the yield of archaeal and bacterial ammonia and nitrite oxidation is high enough to sustain such a complicated food web. The yield of ammonia-oxidizing bacteria is around 0.07 mol C per mol ammonium oxidized and the yield of ammonia-oxidizing archaea is around 0.08-0.09 mol C per mol ammonium oxidized. I would also like to point out that these are retrieved using laboratory conditions under optimal growth conditions, which is never the case in the environment. How high must the ammonia flux into these systems be to fuel such a food web? It would have been great if the authors measured more ammonia concentrations and calculated an ammonia flux. This way we could have at least determined whether ammonia (or nitrite) oxidation could at all fuel such a high carbon fixation rate. However, with the current data set this is not possible.
- 4) The authors discuss consumption of sulfides and sulfites. I also find this difficult to grasp. First of all, both of these compounds have a very short life time as they react very rapidly (chemically) with dissolved oxygen. Then, which microorganisms would be producing these sulfides and sulfates? These sulfur compounds could emerge from sulfate reduction, but for this reaction to occur electron donors are necessary (8 e⁻ per reaction). It is unclear to me where these electron donors would emerge from. Further, it is unclear to me what this has to do with the main message of the manuscript.
- 5) The authors are discussing the consumption of complex organic matter. This topic comes pretty much out of the blue, and also somewhat contradictory to the main message of the manuscript. If as the authors claim complex organic matter of allochthonous origin can come into the RIS, how can they also claim that this system is running on organic carbon created by dark carbon fixation by ammonia and nitrite oxidizers?
- 6) The authors are not consistent in their use of nitrification. At times they mean ammonia oxidation, nitrite oxidation or both. Please just use ammonia oxidation, nitrite oxidation, ammonia and nitrite oxidation when these reactions are meant. This would prevent confusion at many places in the manuscript.
- 7) The authors are not consistent in their use of transcription and expression. At times they use the term "expressed" where it appears that they are referring to transcripts. This is not only incorrect, but also very confusing especially considering the fact that the authors have actually also measured expression.
- 8) It is not correct that "metabolic genes" are generally single copy genes. To name a few amo, hao, nxr genes are a lot of the times found in multiple copies.
- 9) The authors state that they have identified "determinants" of 4-hydroxybutyrate cycle and reductive TCA cycle. Many enzymes involved in these C fixation pathways (especially the former) are bidirectional and also used in other cellular processes. Therefore, I was wondering how were the authors sure that these enzymes were involved in C fixation, but not used in another pathway or which directions these enzymes were operating in?
- 10) Line 135 - it is unclear what the authors mean by turnover time. Do the authors mean doubling time for microorganisms?
- 11) Line 150-152 - It is not clear to me how one can circumvent biases in a situation where the outcome is uncertain.
- 12) Line 188 - energy conservation

- 13) Line 206 – 208 – this could just as well be a problem of the detection limit of the used methodology
- 14) Line 257 – I suppose you mean motility?
- 15) Lines 267- 268. If it is lateral advection, where is this carbon coming from?
- 16) Line 265 – 280 this section is very difficult to understand. For example, what is meant with heterotrophic production? This is almost an oxymoron. The authors actually mean heterotrophic growth, ie conversion of already existing organic matter into cellular matter. This is only a reaction that is recycling already existing carbon in the system. Further, if this is the case that heterotrophic growth is so low in this system, how can this be reconciled with statements in lines 302- 315; e.g., how could these communities adapt to degrading complex organic compounds? Why is leucine incorporation used for determining heterotrophic growth? Not all heterotrophs take up leucine and they do not necessarily take it up at the same rate.
- 17) Line 322- “life at lowest possible seawater temperatures”. Do you mean before seawater freezes? But there is also life found in frozen seawater. This statement does not make sense.
- 18) Lines 325-327 These statements cannot be made without direct measurement of nutrients from many more samples.
- 19) Lines 327-329 ocean ice melting comes out of the blue.

Reviewer #3 (Remarks to the Author):

The manuscript “Lifting the lid: phylogenetically and functionally diverse microorganisms reside under the Ross Ice Shelf” is the most recent and thorough description of microbial life under the Ross Ice Shelf to date. It is the first study to use the full suite of molecular tools developed over the last couple of decades to study microbial communities and the authors do an overall excellent job describing the dominant microorganisms and biogeochemical processes that sustain life in this dark, cold, and relatively isolated environment. The manuscript uses a significant amount of microbial data, in the form of DNA, RNA, and protein sequence data, to draw conclusions about the roles of different organisms beneath the ice and the overall processes underlying shifts in community structure. The manuscript will make a valuable contribution to the literature. I do have some questions regarding the interpretation, not so much of the data as a whole, but of the support for three distinct communities.

16S data: Evidence that the RIS microbial communities are more similar to polar and bathypelagic microbial communities than they are to other marine environments is clearly indicated in multiple panels in Figure 2. Can the author’s please expand on the evidence indicating that the microbial communities from 400, 550, and 700m are distinct from each other (as noted later in Figure 6 and in some of the supplemental data). The communities look very similar in Fig. 2b, even if there are statistically significant differences. Do the branches in the below-RIS group together according to these three categories or some combination of them, like 400m form one cluster and 550 and 700m form another? The description in lines 156-160 and in other places could be clarified. As written, here and in the figure legend, the Authors are comparing RIS to deep ocean globally, not sub-ice-RIS to deep ocean-RIS. Is that correct? Are the differences between miTAGs and amplicons for each sample type (Figure S2 C) as significant as the differences between each sample type (depth)? Can the authors add the same miTAG plots to Figure S3 (a and b) and if so, do they support the same conclusion as the amplicon plots?

MAGs and SAGs: The tree provides a lot of information that is not really needed. Can the same information be represented in a tree with wedges for each phylum, the size of which corresponds to the total number of MAGs and SAGs used? All the details with the appropriate accession numbers are listed in the supplemental data. Are these genomes representative of the communities indicated by 16S rRNA (miTAG and amplicon) analyses? Do they indicate the same differences, in terms of relative abundance, in the major taxa at each depth?

Relative abundance of genes and transcripts: The differences in gene and gene expression profiles for each sample type look more similar than they do different (Fig. 4 and 5). The differences in CAZyme look a little more convincing, but overall it seems like everything is everywhere and the dominant functions and taxa remain the dominant functions and taxa. In the Figure 5 legend –

What are “the top 50 reconstructed genomes”? Is this most complete genomes, the most genes, the most transcripts, or the most CAZymes?

Sparse data: Overall I like the schematic illustration of the major microbial groups and the key processes, but what is the evidence for sulfur oxidizers or sulfur oxidation as a major process that is more significant at depth. There don't appear to be any differences in Fig. 4. It is mentioned in the abstract, on line 224, and in the Figure 6 legend. Figure S10 also has a tree of the taxa. No statistics are provided in Table S10. The virus sequence data and metaproteomic data are not particularly robust and don't add much to the story. 121 proteins detected in the metaproteomic dataset is small, especially for 9 samples. How were triplicates treated? I assume everything was combined because of the low number of IDs?

Overall, the data strongly support the existence of a unique and interesting below-RIS community. While there is evidence that there is more ammonia near the ice and recalcitrant organic matter entering near the seafloor, I am not as convinced that the microbial communities and functions match these differences, at least not in a stark and clearly defined way. Key functions and groups were identified throughout. For example, ammonia and nitrite oxidation were among the dominant transcripts detected at each depth (Fig. 4). Is it more important to point that these functions were more highly expressed in the surface relative to depth or that they were among the most highly expressed functions everywhere below the ice? The text is careful not to overstate this, but I find figure 6 a little stark. Maybe showing all the processes at each depth in Figure 6, but using symbol sizes to indicate enhanced activity. I guess I feel like it's equally interesting and important to show how similar the ecosystem is.

Reviewer #4 (Remarks to the Author):

The current study presents data collected from beneath the Ross Ice Shelf through a borehole drilled using hot water, approximately 300 km from the open waters of the Ross Sea. The study focused on samples collected from three depths beneath the ~370 m thick ice sheet. The data presented are unique – very few datapoints from beneath ice shelves exist, and even fewer have used molecular techniques to so completely characterize the ecology. The paper is generally well-written, but the conclusions drawn are not always sufficiently discussed and supported. I attempted to provide detailed remarks to help in this regard, and I do think the paper is important and interesting. I have a few methodological concerns as well, which are detailed below, and I hope can be easily addressed.

Specific Comments:

Line 71: I think the point you are trying to make is that water samples have been collected only a few times from the ice shelf interior; however, as written, the introduction should also acknowledge the work of Begeman et al at the RIS grounding zone.

Begeman, C. B., et al., . 2018. Ocean Stratification and Low Melt Rates at the Ross Ice Shelf Grounding Zone. *J. Geophys. Res. Ocean.* 123: 7438–7452. doi:10.1029/2018JC013987

It would also make sense to use refs 3-5 to provide true context for the current study (after all, they are the source waters) rather than dismissing them as only accessing the margins. The current paper would be improved by clearer integration with existing knowledge, and this would not detract from the uniqueness and importance of this dataset.

Line 101-102: Water column depth, and sampling depths, are usually described as meters below the bottom of the ice, rather than meters below the ice surface. I would suggest modifying the way that depths are referenced in order to make the work more easily comparable to other studies. I found myself having to repeatedly remind myself as I read that “400 m” is a near-ice-water-interface sample.

Line 103-107: Can these different layers be identified as known water masses (e.g. high salinity shelf waters or ice shelf waters)? This should be possible based on temperature/salinity data and would make the study more directly comparable to other work. If the identification of water

masses is not possible, please describe the layers within the context of known oceanic structure. I note that water masses are identified in reference 13 and that this information should be directly applicable to the current paper.

Line 112: "Nutrient concentrations beneath the center of the RIS are..." should be modified to the past tense, when referring to data in this paper. Please check that this is corrected throughout the paper (I note another instance at line 114).

Line 125 – 128: The argument that ice-shelf basal melt contributes ammonium would be strengthened if average concentrations for the Ross Sea were also reported for context. The references (17 and 18) also note that sea ice is a source of ammonium, however, the ice here is glacial ice, so may not be directly comparable. Please address the expected differences between the types. Further, ref 13 provides melt rates and other data that should make it possible to calculate what the ammonium concentration of the melt water would have to be in order to account for the observed ammonium enrichment. Such a calculation would make the conclusion seem more plausible, if it supported the idea that melt could supply sufficient ammonium. Along the same lines – how is the ammonium getting into the bottom of the ice shelf? I believe there is a rationale for the high ammonium concentrations in sea ice, but does something like this exist for glacier ice? I am not sure that there is a convincing reason to assume that ice originating from the Antarctic Ice Sheets should contain significant ammonium.

Line 133: I do not believe the acronym "PHP" was previously described. Please spell out on first use.

Line 165: Please provide more information on the indicator species analysis, including how the signature species are derived, and what parameters were used in the analysis (for example, I believe you can select whether to limit the analysis to only species that represent individual sites, or to allow groups of sites, and this changes the calculations). Where signature species determined for all of the open and sub-ice communities? It is not clear what the signature species noted are signatures of.

All statistical methods used should also be described in the "Methods" section.

Line 168: If the signature species represent 9% of the RIS community, how do they compare to the open ocean samples?

Line 191: In consideration of the general readership of this journal (rather than microbial specialists), please identify whether organisms are members of the Bacteria or Archaea (e.g. Nitrosphaerales). Also, I note that reference 3 also found abundant Nitrosopumilus and putatively ammonia-oxidizing archaea in deep water. If the current study referenced known water masses, it might be possible to compare these data (or other relevant data) more directly.

Line 200-202: The sentence, "In line with an autotrophic nitrifying bacteria, we identified the determinants of ammonium- or nitrite-dependent carbon fixation via the archaeal 4-hydroxybutyrate cycle and Nitrospina reductive tricarboxylic acid cycle" is confusing. It is not clear how an archaeal C fixation pathway is "in line" with autotrophic nitrifying bacteria. Please consider rewording to make the point clear.

Line 203: It would be helpful to remind the reader how far beneath the ice-water interface 400 m is.

Line 265: The statement at the beginning of the discussion implies that the samples collected from this single borehole represent the entire RIS microbiome. I suggest rephrasing such that the inference is more reflective of the scope of the samples reported here.

Line 266: delete "uniquely"

Line 273: What is meant by "net nutrient consumption"? Nutrients are not consumed, and their use does not result in automatic removal from the system – nutrients are transformed and may

enter different parts of a system, depending on how they are transformed. Therefore, they have to go somewhere. Where are they going, and in what form, in this system, and how does that explain lower concentrations farther under the ice shelf?

Line 286-287: The coherence between dark C-fixation and organic carbon demand suggests that it is possible that the ecosystem beneath the RIS is largely sustained by dark C-fixation, but does not indicate that with certainty. A more complete evaluation of this argument would require discussion of organic carbon concentrations and potential sources of DOC in the sub-ice environment. Further, the metagenomic data reported suggest that organisms in the deep waters are focused on breaking down recalcitrant material, not freshly produced, labile DOC. A more complete evaluation should also include this in the discussion.

Line 295: "Antarctic aquatic environments" implies consideration of both freshwater and seawater (with good reason – it is indeed interesting that this process seems to be important under ice in general); if this is the intention, I would suggest citing a freshwater paper that also shows this such as work from Subglacial Lake Whillans (e.g. Achberger, A. M., B. Christner, A. B. Michaud, J. C. Priscu, M. L. Skidmore, and T. J. Vick-Majors. 2016. Microbial community structure of Subglacial Lake Whillans, West Antarctica. *Front. Microbiol.* 7: 1–13. doi:10.3389/fmicb.2016.01457 or Vick-Majors, T. J., A. C. Mitchell, A. M. Achberger, and others. 2016. Physiological ecology of microorganisms in Subglacial Lake Whillans. *Front. Microbiol.* 7: 1–16. doi:10.3389/fmicb.2016.01705)

Alternatively, if you prefer to maintain a marine focus, you could change "aquatic" to "marine".

Line 296: change "sheets" to "shelves", as refs 9 and 10 both refer to ice shelves, not sheets.

Line 300: please clarify what is meant by "basal layer".

Line 301: The mention of "exceptionally high ammonium concentrations" made me wonder whether the ammonia-oxidizers detected are usually associated with high ammonium concentrations, or low ones? How do they compare to ammonia oxidizers found in lower ammonia sub-ice waters, such as in ref 3, or elsewhere in the region? Inclusion of this point in the discussion could bolster claims about the ammonium concentrations.

Line 308: How is "energy limitation" being defined? An abundance of ammonium is described, which would serve as energy source for abundant community members. No organic C data are provided with which to assess potential energy limitation of heterotrophs. I suggest carefully evaluating what is meant by energy limitation, and considering the large body of literature that exists on that topic.

Line 309: "lithoautotrophic primary producers" – the paper mostly focused on ammonium oxidizers – are others present?

Line 310: delete "slowly", as it is subjective.

Line 311: I assume that "into the shelf" here means "under the ice shelf". Please reword for clarity.

Line 312: Similarly, please use consistent terminology, e.g. "below ice-shelf". Note also that much literature (including ref 13, which provides physical data for this site) refers to below ice shelf waters as "cavities". The authors may wish to consider modifying their terminology in order to remain consistent with the existing literature on these types of systems. Please take note of this throughout the paper, but I especially noticed references to "shelf" throughout the discussion.

Line 312: I do not think that reference 9 reported diatoms in sub-ice shelf waters. Horrigan et al found "virtually undetectable chlorophyll-a". Perhaps you were thinking of Holm-Hansen et al., 1978. *Microbial life beneath the Ross Ice Shelf. Antarct. J. United States* 4: 129–130

which I believe did report a few dinoflagellates and some diatoms at J9. Your ref 3 also reported phototrophs dominated by *P. antarctica* and some diatoms under the McMurdo Ice Shelf, which is

also consistent with water advected from the open Ross Sea and estimated a 10 year decomposition time. These support your contention that some photosynthetically produced organic matter might be important under the ice.

In addition, I would like to point out that the discussion of the eukaryotic community detected was minimal and suggested the presence of mainly heterotrophic eukaryotes (e.g. line 143). The supplementary data presented are not really sufficient for the reader to draw real conclusions about what was detected. So, if an argument about phytoplankton is going to be made, a more complete discussion of the eukaryotes (or at least the potential phototrophs) should be included.

Line 317: "carbon-limitation" cannot be substantiated, as carbon was not measured, nor were potential sources of carbon comprehensively addressed.

Line 320-322: This argument is a stretch – the trace gas supported life was clearly demonstrated, whereas the work shown here (while important and interesting!) did not demonstrate anything about "minimal nutritional requirements". In fact, the paper argues the opposite: that this is a nitrogen rich oasis fed by ice-melt.

Line 329-332: Beginning with "Assuming that", this is only true if ammonium is indeed coming from the ice shelf, which hasn't been convincingly argued for, and if ammonium oxidation is indeed the most important autotrophic process beneath the ice. I suggest reconsidering this statement. But, I completely agree with the next sentence: baseline data such as these are extremely important.

Line 337: please state distance from open water in the site description.

Line 354-356: A convincing argument for lack of contamination would show the referenced salinity, temperature, depth profiles. Similarly, the clearest way to demonstrate that contamination was prevented would be to include the control samples in the MDS plot in S3b.

Ca Line 361 onwards: regarding the liquid water that was removed from the WTS for determination of inorganic nutrient concentrations: were controls run to show that this method did not contaminate the samples with, for example, ammonium? Ammonium is easily contaminated. Perhaps this is not a problem, but given the arguments made for high ammonium concentrations, contamination prevention methods should be clearly detailed.

Line 375-379: These samples should be shown in S3b so that their relationship to the water samples can be easily seen.

Line 405: PHP samples are not usually stored frozen because freeze-thaw can damage cells, causing loss of incorporated radioactivity, and leading to diminished detected rates. Please address this.

Line 413: Please add a citation for the activation energy used.

Line 385-388: This seems in contrast with above description of water sample collection for inorganic chemistry from the WTS. Please clarify.

“Lifting the lid: phylogenetically and functionally diverse microorganisms reside under the Ross Ice Shelf” - - point-by-point response to reviewers’ comments.

Dear reviewers,

First, we want to thank you for evaluating our work and for the constructive and helpful suggestions and comments on the manuscript. We modified the manuscript according to your suggestions and hope that you find that the changes sufficiently address your concerns.

Following the suggestion by reviewer 4, we have modified the description of the sampling depths, as measured from the ice-sea interface (and not from the top of the ice shelf, as they were originally). These correspond now to 30 m, 180 m, and 330 m below the bottom of the ice shelf.

Below, we provide a point-by-point reply to all comments (in blue, and italic).

The line numbers refer to the document with marked changes.

Reviewer #1 (Remarks to the Author):

In this manuscript the authors conduct the first in-depth characterization of microbial community structure and function below the Ross Ice Shelf. Several techniques were applied including metagenomics, single-celled genomics, metaproteomics, metatranscriptomics, and amplicon sequencing. They identified a community below the RIS that is distinct from other polar seawater communities and deep ocean samples. This work represents an important contribution to our understanding of microbial diversity and biogeochemistry beneath Antarctic ice shelves, an expansive yet understudied environment.

General comments:

*The statistics are presented in a somewhat haphazard manner and insufficiently described in the methods. For example, the p-values are often presented as ($p < 0.05$) instead of exact values,

We have moved the statistical approaches (originally in Supplementary Methods) to the Methods Section and have provided more details on the analyses (lines 661-716).

The exact p-values for the IndVal analyses were available in Supplementary Table 1 in the submitted manuscript. We have also added the exact p-values for the univariate statistical comparisons (using Kruskal-Wallis test) to Supplementary Table 1. Following the reviewer’s suggestion, we have also now included the exact p-values when these analyses are introduced in the text: e.g., line 189; lines 199-200, and Supplementary Discussion.

and the indicator species analysis is not described in the methods.

Due to space constraints, we had described all statistical approaches, including indicator species analysis, in the Supplementary Methods. We have moved this Methods section to the Main Text and added details to the methodology (lines 659-713).

*Were p-values corrected for multiple comparisons?

For the GLM approach, univariate tests were performed on each taxon and corrected for multiple comparisons using the $p.uni="adjusted"$ in the anova function. This is specified in the Methods section (line 709).

In the case of Indicator species analysis, each IndVal analysis is conducted for each species independently. We are aware that there are associated multiple-testing issues with this procedure, especially when reporting the number of indicator species per site (De Cáceres et al, 2010), and that such statements should correct for multiple testing by adjusting the p-values.

However, in our study, we are mostly interested in presenting certain individual species as indicators, as reflected in the text: "... identified 'signature species' of the ocean cavity (with respect to deep open-water polar communities) belonging to the phyla PAUC34f, Planctomycetota, and SAR324..."

In this case, we are not making an experiment-wise statement and, as de Cáceres also suggests, there is no need to do any p-value correction.

*Because the proteomic and metatranscriptomic data aren't really being compared to anything (i.e. there is no control) these measures are of minimal value. They don't seem central to the manuscript, and the proteomics data is hardly mentioned. I recommend focusing this manuscript on the amplicons, MAGs, SAGs, which is certainly enough data and quite novel given the environment.

As is the case of many environmental metaproteomic studies, it is not possible to determine a baseline state of the ecosystem to use as control. Contrary to experiment-based approaches, the purpose here is not to compare protein expression in different conditions, but to corroborate the inferences made from the metagenomic and metatranscriptomics data.

In any case, we agree with the reviewer that the metaproteome dataset is relatively small to draw robust conclusions, and we have removed it from this study.

This is not the case of metatranscriptomic data, which we do consider central to the manuscript and suggest maintaining. Its purpose is to demonstrate that community members are not just present, but also transcriptionally active. Whereas the metagenome is shaped by evolution, the metatranscriptome is a snapshot of currently expressed genes at the time of sampling. In particular, we were able to describe the abundance to activity ratio, key metabolic pathways being transcribed, and the differential expression of reconstructed genomes across depths. We believe that this information is important to understand which microbes are active beneath the RIS, to substantiate the genomic findings, and it should thus be maintained in the manuscript.

*The best external points of comparison are likely to be Antarctic winter samples. I believe all the cited studies are from summer. I'm not aware of any modern metagenome studies for winter, but there are some amplicon studies that include winter or early spring samples for the Antarctic (including from the Ross Sea) and these should be brought in.

We agree that including earlier observations of seasonal variability of Antarctic communities is a valuable point of discussion and we have added a paragraph describing the importance of dark carbon fixation in polar environments during winter (line 359-368).

While it is interesting to observe that the same processes that drive primary production beneath the shelf appear are present during the dark polar winter, these seasonal shifts in community composition have been predominately studied in surface waters. It is hard to assess whether the communities in the deep (and permanently dark) water column will experience such shifts.

Therefore, we believe that mesopelagic and bathypelagic polar waters are good external points of comparison, as the cluster analysis in Fig 2c shows. It has to be noted from this analysis, that some polar surface water samples of the poles do cluster with other deep water samples and the below-RIS community. We can only speculate as to why this is, since the season when these samples were collected is unfortunately not available in the metadata of the respective studies (Zhang et al, 2020).

Finally, we must decline the suggestion of the reviewer to include amplicon sequencing studies to the metagenomic comparisons. We decided to limit study comparisons to metagenomic datasets only, due to biases in PCR-derived data (please see answers to Reviewer 2 (comment 11) on a related topic below).

*No data is available yet at either of the cited BioProjects. Please be sure to include your data so that this can be verified prior to publication. Also, recommend pointing the reader to NCBI BioProject database in case they don't recognize the origin of those project numbers. The data cited was unavailable to the public due to moratorium.

We have verified that the raw reads, assemblies and genomes are fully accessible and available to the public under the Bioproject numbers PRJEB35712 and PRJNA593264. Following the reviewer's recommendations, we have explicitly referred to the accession numbers as part of the BioProject database of both NCBI and ENA (lines 1158-1159).

Specific comments:

*Line 163 - here and elsewhere, be cautious about characterizing something as "abundant" when you mean relative abundance. Of course since you have abundance data for RIS and open water I encourage you to make the conversion to absolute abundance where appropriate to do so.

For clarity purposes, we have opted to not use conversions to absolute abundance values in this study. Following the reviewer's suggestion, we have specified in all statements that "abundance" data means "relative abundance".

Line 206 - Is this remarkable? Isn't gene expression typically dominated by a small but variable fraction of the genome? If anything 25 % seems rather high.

Given the potential source of confusion in this statement, we have rephrased as follows:

"Consistent with these results, reconstructed genomes from the genera Nitrosopumilus and Nitrospina were among those with highest relative transcriptional activity in our dataset (S4, S5). These groups express a small fraction of their genomes (i.e. ~25% of total genes at 30 m) compared to other community members (Fig S4d-f), devoting most of their transcriptional effort to the key processes of carbon fixation and ammonia and nitrite oxidation, respectively."

We have also added a new set of plots to Fig S4, where the gene expression of all reconstructed genomes can be compared. As this figure shows, there is a continuum of expression among all genomes, ranging from near-zero values to close-to 100%. We have also added this data in Supplementary Table 1.

The percentage of genes expressed, or diversity of a genome's transcriptome, has been used in similar studies combining genomic and expression data (e.g. (Gifford et al, 2013). This parameter is also connected with the concept of "environmental super-niche" in microbial communities, a term to define divergent adaptive strategies among microbial community members (defined by e.g. (Polz et al, 2006). With respect to this ecological interpretation, we have added a paragraph to this in the discussion as well (lines 408-420).

Line 255 - Proteins don't get highly expressed, genes do.

We have corrected "Proteins..." to "Genes encoding for [...] were highly expressed"

Why would cold-shock genes be expressed in such a homogenous environment?

As reviewed by (Barria et al, 2013), the degree of adaptations to cold stress varies in different bacterial groups. For mesophilic bacterial strains (such as E. coli), cold-shock chaperones, such as cspA, cspB, cspG have been reported to be transiently expressed during cold adaptation (Phadtare, 2012). However, in psychrophiles, there is evidence for constitutive expression of cspA and other molecular chaperones (see (Kumar et al, 2020) and (Raymond-Bouchard & Whyte, 2017) and references therein).

Secondary structures within mRNAs are more likely to form as extracellular temperatures decrease (Ermolenko & Makhatadze, 2002), and thus can reduce transcriptional and translational efficiency. Therefore, it has been suggested that cold-adapted microorganisms activate these cold-shock network to maintain the protein homeostasis which protects cold-induced protein misfolding, especially in a homogeneous, permanently cold environment.

We have added similar explanatory sentences to the text for clarification (lines 305-319).

Line 311 - Given the metabolic plasticity of dinoflagellates it seems odd to invoke photosynthesis here, seems far more likely that they are living heterotrophically, regardless of whether endemic or transient.

We agree and have limited this discussion to members of the class Bacillariophyceae, which are almost exclusively phototrophs, and had previously been observed below the RIS.

Line 327 - Remove "of" near end of this line.

This has been corrected.

Line 411 - This estimate from activation energy requires some justification/explanation.

The value for activation energy was empirically determined for bacterial heterotrophic production in the mesopelagic ocean (Lønborg et al, 2016). This study analysed the temperature dependence of PHP using an Arrhenius type relationship and observed that the resultant apparent activation energies (E_a) were fundamentally different between the epi-, meso-, and bathypelagic ocean layers. In the mesopelagic zone the E_a of $72 \pm 15 \text{ kJ mol}^{-1}$ was very close to the theoretical value predicted for overall heterotrophic organism metabolism, suggesting that factors other than temperature (e.g., substrate) also limit the PHP in this layer (Brown et al, 2004; López-Urrutia & Morán, 2007) and indicating that temperature is one of the main limiting factors for the PHP in this layer.

For clarification, we have added the citation (Lønborg et al., 2016) to the Methods section, now line 565).

Line 440 - Those primers have been superseded by others that are more sensitive to SAR11 and archaea (both important for this study) (e.g., Walters et al. 2015, mSystems). Please double check that Caporaso et al. 2012 was in fact the set used, and justify/provide caveats if so.

We agree, this is an error in citation. The correct reference for Earth Microbiome Project barcoded primer set and conditions is the following: (Thompson et al, 2017).

We have changed the reference accordingly and specified the corrections on the original primers to fix known biases against SAR11 and archaea (lines 591-597).

Reviewer #2 (Remarks to the Author):

The current manuscript by Martínez-Pérez et al. uses combined omics techniques to study the microbial diversity present under Ross Ice Shelf (RIS). I would like to state that I appreciate the difficulties associated with acquiring and analyses of samples originating from such remote places. I also find the topic potentially interesting; however, the manuscript has many drawbacks that I do not believe can easily be remedied. Unfortunately, the authors do not provide convincing (or bluntly any) evidence that substantiates their supposition that ammonia oxidation and nitrite oxidation derived carbon fixation fuels the dark microbial food web. This supposition is indeed the main thread throughout the manuscript; however, there are several side lines that pop up throughout the manuscript without much of an explanation and connection to the rest of the manuscript, for example consumption of sulfur compounds, recycling of complex organic material or ocean warming. As such, these side lines do not make the manuscript stronger, but very confusing to go

through as if the authors could not agree upon what to include in the manuscript.

Comments:

1) The authors use only three sampling points in their nutrient depth profile. This is incredibly limited and cannot be used to pinpoint any microbial activity to any zone in the depth profile. In order to achieve this a much higher resolution nutrient profile would need to be determined.

We should note that drilling through the ice shelf and sampling the water column beneath is extremely challenging, especially at the remote location where our sampling site was located (~300 km from the shore). It is worthwhile reiterating that this location was not accessed in the past 40 years for this reason. In these conditions we prioritized a sampling strategy that would ensure minimum disturbance and contamination (i.e., using an in-situ pump to collect seawater samples), at the cost of being time-intensive and preventing additional sampling depths.

As a result, we agree that this study lacks the necessary sampling resolution to draw conclusions about the depth-related differences observed in community composition, metabolic capabilities, and gene expression. Therefore, we have removed these observations from the main text. However, given the depth-related differences are striking and hinted at in the figures, we have opted to mention them in Supplementary Discussion, acknowledging that additional depth profiles would be necessary to confirm these results.

2) The authors suggest that the highest dark carbon fixation rates are at the sea-ice boundary layer. They infer this based on the high abundance of ammonia-oxidizing microorganisms in this sampling point and that the highest measured ammonium concentrations were here. These measurements and experiments cannot be used to determine ammonia and nitrite oxidation rates or carbon fixation rates. In order to bring the authors' assertions to the level of hard evidence-based facts, ammonia oxidation rates and carbon fixation rates must be measured. The authors should keep in mind that per cell activity is not a constant thing in a culture, let alone in the environment. Furthermore, it depends on many environmental factors ranging from availability of resources to the presence of predators. Therefore, for studies such as these it is paramount to provide direct measurements of the targeted activities rather than just proxies for these activities.

We are aware of the limitations of omics approaches and have worded our conclusions very carefully. In particular, we don't only refer to the presence and abundance of ammonia-oxidizing organisms, but also refer to the expression of genes responsible for autotrophy in the nitrifying community.

We agree that transcriptional activity is not a direct measure of activity rates, but we base our conclusions in previous studies of under-shelf environments, where nitrification and/or carbon fixation rates have been quantified. Since these past studies lack detailed compositional data, the aim of this study was mainly to identify the distinct community members under the RIS relative to those in the open ocean areas, confirm their metabolic capabilities, and infer their ecological relationships. We identified the mediators of nitrification and detected other facultative chemolithoautotrophs that likely contribute to dark fixation beneath the shelves. Paired with the previous activity data, our study highlights that ocean cavity waters are primarily chemosynthetically-driven systems.

Finally, while providing in-situ rates to this study would have been desirable, it lays beyond the scope of this study. We must reiterate the difficulties in accessing this environment, the time-limitations on site, and the obstacles to be granted authorised permission to perform certain stable or radioactive isotope incubations in situ.

3) I am not convinced that the yield of archaeal and bacterial ammonia and nitrite oxidation is high enough to sustain such a complicated food web. The yield of ammonia-oxidizing bacteria is around

0.07 mol C per mol ammonium oxidized and the yield of ammonia-oxidizing archaea is around 0.08-0.09 mol C per mol ammonium oxidized. I would also like to point out that these are retrieved using laboratory conditions under optimal growth conditions, which is never the case in the environment. How high must the ammonia flux into these systems be to fuel such a food web? It would have been great if the authors measured more ammonia concentrations and calculated an ammonia flux. This way we could have at least determined whether ammonia (or nitrite) oxidation could at all fuel such a high carbon fixation rate. However, with the current data set this is not possible.

As the reviewer indicates, there are insufficient data points to accurately calculate ammonium fluxes under the RIS. With the available data, however, we can still estimate an upper boundary for carbon fixation under the RIS, using the measured ammonium concentrations in the water column. One can calculate the rate of ammonium oxidation at each depth, based on the measured ammonium concentrations, and the specific nitrification rate (λ_{nitrif} , the daily rate of ammonium oxidation divided by the corresponding ammonium concentration). As a representative value, we used the world median λ_{nitrif} , 0.195 d^{-1} (Yool et al, 2007). The calculated ammonium oxidation rates are listed in the table below (column 4).

Water-column regions	Depth [m]	NH_4^+ [μM]	NH_4^+ oxidation [$\mu\text{M NH}_4^+ \text{ d}^{-1}$]	Substrate use efficiency [C fixation/ NH_4^+ oxidation]	AOA-fueled C fixation [$\mu\text{M C m}^{-3} \text{ d}^{-1}$]
IBL	30	0.44	0.09	0.09	8.1
V-IL	180	0.05	0.01	0.09	0.9
S-IL	330	0.04	0.01	0.09	0.9

The resulting ammonium oxidation values in the IBL ($90 \text{ nM NH}_4^+ \text{ d}^{-1}$) are in accordance to rates measured in Southern Ocean waters with comparable ammonium concentrations (e.g., AASW with mean $0.7 \mu\text{M NH}_4^+$ support a mean ammonium oxidation of $62 \text{ nM NH}_4^+ \text{ d}^{-1}$ (Tolar et al, 2016b). Unfortunately, no carbon fixation rates were provided in the cited study.

The conversion to carbon fixation can be calculated based on experimentally defined substrate use efficiency, as the reviewer pointed out. At a value of $0.09 \text{ mol C fixed per mol NH}_4^+$ (Berg et al, 2015), the community in the IBL has the capacity to fix up to $8.1 \mu\text{mol C m}^{-3} \text{ d}^{-1}$. This is a value very close to that measured by Horrigan and colleagues (Horrigan, 1981) beneath the J9 borehole ($8.3 \mu\text{mol C m}^{-3} \text{ d}^{-1}$), and a value in accordance to the total carbon demand under the RIS ($6\text{-}12 \mu\text{mol C m}^{-3} \text{ d}^{-1}$), estimated in this study from heterotrophic production rates (Table 1 & line 343)

Finally, we must note that the above calculations have not considered the second step of nitrification (i.e., nitrite oxidation) and thus a fraction of ammonium-fuelled autotrophy is not accounted for. Therefore, even if these calculations can be considered an upper-end for AOA-fuelled autotrophy, we can conclude that the measured ammonium concentrations can realistically support a significant fraction of the total dark carbon fixation under the RIS. As a comparison to existing environmental measurements, the ammonium concentration measured at 200m beneath borehole J9, (the depth from where the C-fixation rates were detected (Horrigan, 1981) was of $\sim 0.7 \mu\text{M}$ (slightly higher, but in the range of what this study recorded: $0.4 \mu\text{M}$, Table 1).

Given the importance of these estimations to the understanding of the biogeochemical processes beneath the RIS, we have added them to the main text (lines 369-379) and to a Supplementary Discussion section.

4) The authors discuss consumption of sulfides and sulfites. I also find this difficult to grasp. First of all, both of these compounds have a very short life time as they react very rapidly (chemically) with dissolved oxygen. Then, which microorganisms would be producing these sulfides and sulfates? These sulfur compounds could emerge from sulfate reduction, but for this reaction to occur electron donors are necessary (8 e⁻ per reaction). It is unclear to me where these electron donors would emerge from. Further, it is unclear to me what this has to do with the main message of the manuscript.

The principal purpose of this manuscript is to identify and describe microbial metabolisms that can sustain life in the cavity under the RIS. The system appears to be largely sustained by chemolithoautotrophy. We focus on ammonium and nitrite oxidation since the data suggests that nitrifying organisms are among the most abundant and/or active, but also the most environmentally relevant given they have ample substrate supply available from shelf-sourced ammonium. However, there are other members of the community with potential to contribute to chemolithoautotrophy in the oceanic cavity, and which we consider worth mentioning and discussing.

*By phylogenetic analysis of RuBisCO genes, it appears that chemoautotrophs other than nitrifiers are present, specifically facultative sulfur compound oxidizers (belonging to gammaproteobacterial SUP05 clades). Thus, we discuss the possibility that reduced sulfur compounds could also drive chemoautotrophic metabolisms beneath the RIS. This is corroborated by presence and transcription of genes encoding for the oxidation of reduced sulfur compounds, in particular *sqr*, *r-dsrA*, and *soxB* genes. These genes historically have been characterized to use sulfide and thiosulfate as substrates. Since their discovery, however, it has been shown that some of these enzymes can also be involved in the oxidation of other sulfur compounds. To avoid confusion, we have changed to the generic term “reduced sulfur compounds” in the text. We have also rephrased the paragraph in the results section so that this train of thought is more evident to the reader (lines 266-283).*

The current study does not offer evidence for the origin of reduced sulfur compounds or their stability in an oxic water column. However, from recent findings we now know that chemolithotrophic members of the SUP05 clade both have the genetic potential to respire oxygen and are active in marine oxic environments (Callbeck et al, 2018; Marshall & Morris, 2012). Together with omic-based identification of sulfur compound oxidation genes in marine oxic waters, including coastal surface waters of the Antarctic peninsula (Grzyski et al, 2012; Swan et al, 2011; Williams et al, 2012), these findings indicate that these organisms must be capable of using sulfur compounds that are stable in an oxic water column.

Laboratory and environmental studies have shown the capability of members of the SUP05 clade to store elemental sulfur intracellularly, which are used as a reserve in the absence of external energy sources (Callbeck et al, 2018). Further, cultured SUP05 isolates can use diverse sources of reduced organic and inorganic sulfur (e.g., thiotaurine, thiosulfate) to fuel aerobic carbon fixation, even at submicromolar concentrations of these substrates (Shah et al, 2019). There are diverse sources of dissolved organic sulfur compounds in the marine environment, as recently reviewed (Moran & Durham, 2019), and references therein. These include bacterial degradation of sulfolipids from photosynthetic membranes, viral lysates and zooplankton detritus.

We agree that clarifying the above-mentioned topics will be a valuable addition to the manuscript and we have argued for the potential of sulfur-driven chemoautotrophy below the RIS in the discussion (lines 359-368).

5) The authors are discussing the consumption of complex organic matter. This topic comes pretty much out of the blue, and also somewhat contradictory to the main message of the manuscript. If as the authors claim complex organic matter of allochthonous origin can come into the RIS, how can they also claim that this system is running on organic carbon created by dark carbon fixation by ammonia and nitrite oxidizers?

In our efforts to describe the complex community under the RIS and its metabolic potential, we observed that a large portion of the heterotrophic community members was equipped with a suite of enzymes capable of cleaving complex carbohydrates (like α -N-acetylgalactosaminidase (GH109), α -L-fucosidase (GH29) and α -L-rhamnosidase (GH78)). This suggests that the identified heterotrophs could play a secondary role in the degradation of complex polysaccharides by utilizing partly degraded fragments.

The fact that differentially enriched members under the RIS have the potential to degrade these organic remnants does not contradict our prediction that the system is largely supported by dark organic carbon fixation. On the contrary, it supports the notion that phytoplankton-derived organic exudates are absent from the system and that the organic matter present in the centre of the RIS cavity is likely recalcitrant and difficult to access. As a comparison, other systems largely supported by chemoautotrophy (such as the polar surface waters in austral winter) have also considered heterotrophic consumption of recalcitrant dissolved organic carbon (from the previous summer bloom) to be an additional support for the food web. In both systems, the heterotrophic community is likely to rely on newly produced carbon in situ.

Based on our calculations, the system's carbon demand can indeed be met by dark carbon fixation. However, we cannot rule out the possibility that a fraction of the carbon demand is met by degradation of recalcitrant organic matter. In other words, it would be naive to neglect that a fraction of the carbon demand could be supplied by allochthonous carbon sources. We address this possibility more clearly in the revised manuscript and suggest potential origins of organic substrates beneath the RIS (lines 421-443).

6) The authors are not consistent in their use of nitrification. At times they mean ammonia oxidation, nitrite oxidation or both. Please just use ammonia oxidation, nitrite oxidation, ammonia and nitrite oxidation when these reactions are meant. This would prevent confusion at many places in the manuscript.

We have amended the use of this term as suggested.

7) The authors are not consistent in their use of transcription and expression. At times they use the term "expressed" where it appears that they are referring to transcripts. This is not only incorrect, but also very confusing especially considering the fact that the authors have actually also measured expression.

We have corrected this where applicable.

8) It is not correct that "metabolic genes" are generally single copy genes. To name a few amo, hao, nrx genes are a lot of the times found in multiple copies.

We agree with this remark. As the reviewer has correctly pointed out, these metabolic genes are often present twice or more in a genome. We have now expressed the number of genes as average gene copies per organism (Fig 4). This removes the assumption that the genes are always present as

single-copy genes, and instead is simply the gene's abundance relative to single-copy ribosomal gene abundance (an average copy number per organism of 1).

The caption to Fig 4 reads now: "For metagenome reads, the heatmap shows the abundance of each pathway, expressed as average gene copies per organism (across all genes listed in the pathway) calculated relative to the abundance of 14 universal single-copy ribosomal genes."

9) The authors state that they have identified “determinants” of 4-hydroxybutyrate cycle and reductive TCA cycle. Many enzymes involved in these C fixation pathways (especially the former) are bidirectional and also used in other cellular processes. Therefore, I was wondering how were the authors sure that these enzymes were involved in C fixation, but not used in another pathway or which directions these enzymes were operating in?

For clarification, we have now enumerated in the text the marker genes used to identify “archaeal 4-hydroxybutyrate cycle (hbsC, hbsT genes) and Nitrospina reductive tricarboxylic acid cycle (aclB gene)” (lines 246-247). We have carefully curated these databases and homology search cutoffs so that they only capture enzymes that function in the carbon fixation direction. Consistently, our hits for these genes in the MAGs and SAGs are as expected based on current knowledge of the distribution of these two cycles. The detected hbsC and aclB genes affiliate with those of known ammonia-oxidizers and nitrite-oxidizers, respectively.

To validate these findings, this study also provides phylogenetic trees with representative reference sequences of known function (Figs S7-S16, and Supplementary Methods). This is a means to assess the directionality of the reaction encoded. For instance, all dsr genes identified in this study affiliate with known sulfide-oxidising (r-DsrA) clades, from which it can be concluded that the encoded enzymes encode for sulfide oxidation.

10) Line 135 - it is unclear what the authors mean by turnover time. Do the authors mean doubling time for microorganisms?

This is right, the turnover time defines the amount of time required to replace the standing biomass in a system.

We have now specified it as “microbial biomass turnover time”

11) Line 150-152 – It is not clear to me how one can circumvent biases in a situation where the outcome is uncertain.

There is ample evidence describing how PCR-based phylogenetic marker protocols are vulnerable to biases through sample preparation and sequencing errors. For instance, the choice of which regions of the 16S rRNA gene are targeted for sequencing is among the biggest factors underlying technical differences in microbiome composition, (Tremblay et al, 2015).

Avoiding these sources of variability is a means to avoid bias when comparing different studies, and thus a justification of the choice of shotgun metagenomics vs amplicon studies for multi-study comparison.

For clarification, we have included the reference (Tremblay et al, 2015) to the text (now line 178)

12) Line 188 – energy conservation

To correct for the thermodynamically inadequate term “energy generation”, we have specified that by “energy” we refer to “ATP” (implying that these pathways are involved in the generation of ATP, and not simply “energy”)

13) Line 206 – 208 – this could just as well be a problem of the detection limit of the used methodology

We have included a new set of plots to Fig S4, which show the relationship between transcription (TPMs), and percentage of open reading frames (ORFs) transcribed. The genomes form a continuum in the x-axis, where the percent of the genome that is transcribed generally increases with the genome's TPM. This trend could be explained, as the reviewer suggests, with differences in transcript detection for each genome, i.e., a higher diversity of transcripts is more likely to be detected in genomes with high TPMs, and vice versa.

The exceptions to this trend are observed for the genera Nitrosopumilus and Nitrospina (with high TPMs, but low proportion of ORFs transcribed). These genera have already been observed to express a reduced suite of metabolic pathways in environmental communities (Gifford et al, 2013). The "low diversity transcriptome" also conforms to the niche paradigm, by which microbial community members use different adaptative strategies to exploit a particular environment (e.g., Polz et al, 2006; Lauro et al, 2009). As supported by previous evidence and theoretical considerations, we interpret the reduced percentage of ORFs expressed to be an ecological trait of these genera, and not an artefact. We have included these considerations in the revised manuscript, (lines 408-420).

14) Line 257 – I suppose you mean motility?

This was a mistake; we have corrected to motility

15) Lines 267- 268. If it is lateral advection, where is this carbon coming from?

We have clarified as follows:

"...their dependence on in situ chemosynthesis and on lateral advection of food sources from open-water areas"

16) Line 265 – 280 this section is very difficult to understand. For example, what is meant with heterotrophic production? This is almost an oxymoron. The authors actually mean heterotrophic growth, ie conversion of already existing organic matter into cellular matter. This is only a reaction that is recycling already existing carbon in the system.

Heterotrophic production is "secondary production" and represents the formation of living mass of a heterotrophic population or group of populations over some period of. It is the heterotrophic equivalent of net primary production by autotrophs. In microbial assemblages (especially from the marine environment) this term was initially named "bacterial production"; it was shifted to "bacterial heterotrophic production" with the discovery of autotrophic bacteria and to "prokaryotic (heterotrophic) production" to include heterotrophic bacteria and archaea and distinguish from chemoautotrophic production by Archaea. The term has been used in the field of marine microbial ecology for several decades and is a widely used convention. Changing this to "heterotrophic growth", as the reviewer suggests, would be confusing. (For further information, we kindly direct the reviewer to the chapter "Bacterial Production and Biomass in the Oceans" by Hugh Ducklow, in Microbial Ecology of the Oceans, Wiley-Blackwell, (Ducklow, 2000)).

Further, if this is the case that heterotrophic growth is so low in this system, how can this be reconciled with statements in lines 302- 315; e.g., how could these communities adapt to degrading complex organic compounds?

As specified in the text, one of the factors shaping the very low heterotrophic production detected beneath the RIS is the presence of labile dissolved organic matter (DOM). As discussed in the manuscript, phytoplankton is an unlikely contributor to the DOM pool and detrital sources of bacterial substrates may be more important instead. Long-lived DOM might include some polymers (e.g., complex carbohydrates) that are not readily hydrolyzed.

This phenomenon has also been observed in deep-ocean systems which receive little input from fresh, phytoplankton-derived DOM. These environments are characterized by low PHP rates (Baltar et al, 2009), but also by the presence of community members capable of degrading complex carbon sources (e.g. members of the SAR202 clade, which have the genomic potential to degrade aromatic carbon substrates (Landry et al, 2017). In both the deep ocean and the cavity beneath the RIS, the lack of continuous sources or labile organic matter appears to have selected for those organisms capable of using refractory material, thus resulting in an “adapted” community, as observed in increased metabolic repertoire to degrade recalcitrant organic compounds (Aristegui et al, 2009).

Why is leucine incorporation used for determining heterotrophic growth? Not all heterotrophs take up leucine and they do not necessarily take it up at the same rate.

The leucine incorporation technique is used as a method for estimating rates of protein synthesis (Kirchman et al, 1985), which in turn constitutes a significant fraction for total biomass production. The original methodology used leucine as the amino acid of choice since it was observed to be mostly incorporated into amino acids, with a minor fraction being respired during incubation times.

The limitations to this methodology have been assessed and discussed elsewhere (e.g., (Smith & Azam, 1992). Nevertheless, this method has been and is still widely used and accepted by the marine microbial community as measure of heterotrophic production. Importantly, its wide usage over the subsequent decades and across multiple ecosystems has enabled comparisons across studies and time.

17) Line 322- “life at lowest possible seawater temperatures”. Do you mean before seawater freezes? But there is also life found in frozen seawater. This statement does not make sense.

The conditions under the ice shelf are such that they elevate the freezing temperature of the water circulating beneath it, and so the liquid water is colder than the temperature at which it should form ice (reaching values of ~ -2°C). This makes it the coldest “liquid” seawater on Earth. Frozen seawater can be colder than this, as the reviewer points out; however, sea-ice is no longer considered “marine water”, given the absence of dissolved salts and a completely different physical state.

We have removed this exact phrasing from the revised version of the manuscript, which now only includes “...microorganisms under Antarctica’s ice shelves can thrive in some of the coldest [...] marine waters” (line 444-445). We hope this reduces confusion caused by the previous statement.

18) Lines 325-327 These statements cannot be made without direct measurement of nutrients from many more samples.

This paragraph is phrased carefully to indicate to the reader that statements are made with the available results at hand. These are, as the reviewer indicates, limited in spatial resolution, but can be still used to draw conclusions from the studied system.

19) Lines 327-329 ocean ice melting comes out of the blue.

We have stated, at the beginning of the paragraph that ice melting is likely responsible for the presence of high ammonium concentrations beneath the shelf. It then follows that ice melting, susceptible to global temperature changes, might play an important role in shaping the biogeochemistry of sub-shelf systems. We have rephrased this paragraph to connect these statements better (now lines 444-466).

Reviewer #3 (Remarks to the Author):

The manuscript “Lifting the lid: phylogenetically and functionally diverse microorganisms reside under the Ross Ice Shelf” is the most recent and thorough description of microbial life under the Ross Ice Shelf to date. It is the first study to use the full suite of molecular tools developed over the last couple of decades to study microbial communities and the authors do an overall excellent job describing the dominant microorganisms and biogeochemical processes that sustain life in this dark, cold, and relatively isolated environment. The manuscript uses a significant amount of microbial data, in the form of DNA, RNA, and protein sequence data, to draw conclusions about the roles of different organisms beneath the ice and the overall processes underlying shifts in community structure. The manuscript will make a valuable contribution to the literature. I do have some questions regarding the interpretation, not so much of the data as a whole, but of the support for three distinct communities.

16S data: Evidence that the RIS microbial communities are more similar to polar and bathypelagic microbial communities than they are to other marine environments is clearly indicated in multiple panels in Figure 2. Can the author’s please expand on the evidence indicating that the microbial communities from 400, 550, and 700m are distinct from each other (as noted later in Figure 6 and in some of the supplemental data). The communities look very similar in Fig. 2b, even if there are statistically significant differences. Do the branches in the below-RIS group together according to these three categories or some combination of them, like 400m form one cluster and 550 and 700m form another?

We have modified Fig S3, where the differences in microbial communities by depth are compared in separate PCoA plots (based on Bray-Curtis dissimilarity). A PerMANOVA tested whether the differences among sampled depths were significant. Significant differences were observed between depths within each dataset (i.e. within amplicon sequence data and within metagenomic miTags).

The description in lines 156-160 and in other places could be clarified.

We have modified the description here and elsewhere to clarify what environments are compared, i.e. whether the text refers to the community in the ocean cavity under the RIS, or whether we refer to the different depths sampled.

As written, here and in the figure legend, the Authors are comparing RIS to deep ocean globally, not sub-ice-RIS to deep ocean-RIS. Is that correct?

This is correct: the dendrograms in Figure 2 compare samples below the RIS ocean cavity and deep (> 200 m) global samples. For clarification, we have added this to the caption in Figure 2:

“[...] the dashed box highlights the clustering of communities in the ocean cavity under the RIS with global deep-sea environments (in detail in 2d). (d) Heatmap visualization of calculated Z-scores from below-RIS and global deep-sea environments”

And the main text now reads as follows:

“When compared to deep (> 200m) open ocean communities worldwide, compositional differences between open-ocean and below-RIS microbial communities are evident even at the phylum level.” (lines 182-185)

Are the differences between miTAGs and amplicons for each sample type (Figure S2 C) as significant as the differences between each sample type (depth)? Can the authors add the same miTAG plots to Figure S3 (a and b) and if so, do they support the same conclusion as the amplicon plots?

We refer the reviewer to the modified Figure S3, which illustrates the differences between samples, which include both miTag and amplicon datasets.

MAGs and SAGs: The tree provides a lot of information that is not really needed. Can the same information be represented in a tree with wedges for each phylum, the size of which corresponds to the total number of MAGs and SAGs used? All the details with the appropriate accession numbers are listed in the supplemental data.

We have completely revised Figure 3 based on the reviewer's suggestion so it is much more informative and integral to the manuscript. It now provides a visual overview of the MAG and SAG phylogeny in relation to their abundance in the system and metabolic capabilities. It includes information present in the supplementary tables (e.g., genome completeness, CPMs and TPMs), but also the presence of specific metabolic features discussed in the text. Since these characteristics are unique to each reconstructed genome, we have decided against collapsing the tree into phylum-sized wedges.

Are these genomes representative of the communities indicated by 16S rRNA (miTAG and amplicon) analyses?

We have updated Figure 3 with a clearer color-coding, to better indicate that MAGs and SAGs represent most of the taxonomic clades found in the 16S rRNA gene amplicon sequencing data. These include representative genomes of all of the phyla with relative abundance > 0.5% (Figure 2), but also the top four most abundant genera (Figure S2b), including Arctic96Ad-7, Pelagibacter, Nitrosopumilus, and Thioglobus. We have now also explicitly included this in the Results section (lines 221-222):

Do they indicate the same differences, in terms of relative abundance, in the major taxa at each depth?

Figure S6 shows how the relative abundances of genomes from dominant taxa change with depth. These indicate a shift in the community that is in accordance with what was inferred from both miTag and amplicon data. In particular, the relative abundance of Nitrosopumilus is higher in the shallowest depth, whereas the relative abundance of heterotrophic taxa (e.g. Planctomyces, Verrucomicrobia) is higher in the deeper samples

In agreement with Reviewer 2's concerns of a lack of sufficient data to confirm between-depth observations, we have removed discussion of depth-related differences in microbial composition and function from the main text. Instead, we now only note the differences in relative abundance in a Supplementary Comment (see reply to Reviewer 2's comment 1 and related comments below). With respect to the concordance of trends by metagenomes and amplicon data, the comment includes the following:

“The relative abundances of genomes from dominant taxa changed with depth in a concordant trend with miTag and amplicon data (Figure S6). In particular, the relative abundance of the chemoautotrophic genus Nitrosopumilus was higher at 30 m, whereas the relative abundance of heterotrophic taxa (e.g. Planctomyces, Verrucomicrobia) was higher in the deeper samples (Figs. S5, S6).”

Relative abundance of genes and transcripts: The differences in gene and gene expression profiles for each sample type look more similar than they do different (Fig. 4 and 5). The differences in CAZyme look a little more convincing, but overall it seems like everything is everywhere and the dominant functions and taxa remain the dominant functions and taxa.

In the revised version of the manuscript, we have avoided drawing conclusions from the direct comparison of expression profiles of individual genes. We consider these to be relevant

information about the genomic potential expressed by the community but consider erroneous to treat them as differential expression profiles given that normalization (e.g., to genome abundance) is not possible.

As mentioned above, we have also avoided describing depth-related differences in relative abundance between taxonomic groups, as inferred by amplicon sequencing, miTag data analysis, or relative abundance and transcription of reconstructed genomes. However, since some depth-related differences are hinted at in the figures (for instance, relative transcriptional activity of MAGs and SAGs between depths in Fig S4), we have summarized these observations in Supplementary Discussion, acknowledging that additional depth profiles would be necessary to confirm these results.

In the Figure 5 legend – What are “the top 50 reconstructed genomes”? Is this most complete genomes, the most genes, the most transcripts, or the most CAZymes?

We have modified the caption to Figure 5:

“Relative abundance and transcription of selected carbohydrate active enzyme (CAZYme) classes. Data is displayed for reconstructed genomes (MAGs and SAGs) where CAZYme diversity was highest (top 50).”

Sparse data: Overall I like the schematic illustration of the major microbial groups and the key processes, but what is the evidence for sulfur oxidizers or sulfur oxidation as a major process that is more significant at depth. There don't appear to be any differences in Fig. 4. It is mentioned in the abstract, on line 224, and in the Figure 6 legend. Figure S10 also has a tree of the taxa.

The evidence of active sulfur oxidizing organisms, including carbon fixation transcripts from reconstructed genomes, suggests that the oxidation of reduced sulfur species is an additional source of chemoautotrophy. Therefore, we consider their mention relevant to the overall description of the ecosystem beneath the RIS. Our current dataset does not allow us to determine what is the contribution of sulfur compound oxidizers to overall dark carbon fixation in the oceanic cavity. Based on the relative abundance and transcriptional activity of other chemoautotrophs (namely nitrifying taxa), we expected their contribution to be progressively greater with depth, but we have been careful not to overstate this assumption. For instance, no such statement is made in the abstract, nor in line 224 (now 276); and figure 6 depicts the presence of S-oxidizing representatives in all depths.

We have modified Fig 6 to represent all functional groups at all depths, while still representing higher relative abundance and activity of nitrifiers with respect to heterotrophic clades. We understand that the previous representation might have led to confusion and have modified the schematic accordingly (see also last comment to reviewer 3 below)

No statistics are provided in Table S10.

The manuscript does not have a Table S10. Table 1 does offer statistic data (in the form of averages and standard deviations of the samples measured). More statistical analyses are provided in Supplementary Table 1.

We cannot address this particular statement without further detail.

The virus sequence data and metaproteomic data are not particularly robust and don't add much to the story. 121 proteins detected in the metaproteomic dataset is small, especially for 9 samples. How were triplicates treated? I assume everything was combined because of the low number of IDs?

We agree with the reviewer that the metaproteome dataset is relatively small to draw robust conclusions; this observation was also raised by Reviewer 1 and we have decided to remove this

data from the study. We have decided, however, to retain the brief description of the viral component of the community; we consider it does provide an additional layer of knowledge demonstrating the diverse cavity community.

Overall, the data strongly support the existence of a unique and interesting below-RIS community. While there is evidence that there is more ammonia near the ice and recalcitrant organic matter entering near the seafloor, I am not as convinced that the microbial communities and functions match these differences, at least not in a stark and clearly defined way. The key functions and groups were identified throughout. For example, ammonia and nitrite oxidation were among the dominant transcripts detected at each depth (Fig. 4). Is it more important to point that these functions were more highly expressed in the surface relative to depth or that they were among the most highly expressed functions everywhere below the ice? The text is careful not to overstate this, but I find figure 6 a little stark. Maybe showing all the processes at each depth in Figure 6, but using symbol sizes to indicate enhanced activity. I guess I feel like it's equally interesting and important to show how similar the ecosystem is.

We agree with these considerations and have minimized the discussion of in-between depth differences that are fairly subtle or can't be substantiated due to the limited number of samples (see replies above and also reply to Reviewer 2's comment 1). We have explicitly highlighted the presence of dominant functions and groups throughout the water column (e.g. lines 272-273, lines 281-283), and have also modified Figure 6 to reflect this view.

Reviewer #4 (Remarks to the Author):

The current study presents data collected from beneath the Ross Ice Shelf through a borehole drilled using hot water, approximately 300 km from the open waters of the Ross Sea. The study focused on samples collected from three depths beneath the ~370 m thick ice sheet. The data presented are unique – very few datapoints from beneath ice shelves exist, and even fewer have used molecular techniques to so completely characterize the ecology. The paper is generally well-written, but the conclusions drawn are not always sufficiently discussed and supported. I attempted to provide detailed remarks to help in this regard, and I do think the paper is important and interesting. I have a few methodological concerns as well, which are detailed below, and I hope can be easily addressed.

Specific Comments:

Line 71: I think the point you are trying to make is that water samples have been collected only a few times from the ice shelf interior; however, as written, the introduction should also acknowledge the work of Begeman et al at the RIS grounding zone.

Begeman, C. B., et al., . 2018. Ocean Stratification and Low Melt Rates at the Ross Ice Shelf Grounding Zone. *J. Geophys. Res. Ocean.* 123: 7438–7452. doi:10.1029/2018JC013987

It would also make sense to use refs 3-5 to provide true context for the current study (after all, they are the source waters) rather than dismissing them as only accessing the margins. The current paper would be improved by clearer integration with existing knowledge, and this would not detract from the uniqueness and importance of this dataset.

We have re-written parts of the introduction to include previous knowledge of the interior of the RIS (lines 73-82), including the work of Begeman and colleagues on the grounding zone of the RIS (lines 77-78).

Line 101-102: Water column depth, and sampling depths, are usually described as meters below the bottom of the ice, rather than meters below the ice surface. I would suggest modifying the way that depths are referenced in order to make the work more easily comparable to other studies. I found

myself having to repeatedly remind myself as I read that “400 m” is a near-ice-water-interface sample.

We agree with this suggestion and have modified the description of the sampling depths (30m, 180m, and 330m below bottom of the ice) in text, figures, and tables.

Line 103-107: Can these different layers be identified as known water masses (e.g. high salinity shelf waters or ice shelf waters)? This should be possible based on temperature/salinity data and would make the study more directly comparable to other work. If the identification of water masses is not possible, please describe the layers within the context of known oceanic structure. I note that water masses are identified in reference 13 and that this information should be directly applicable to the current paper.

The Reviewer is correct in that the temperature and salinity can be used to identify the water masses. As correctly identified, reference 13 (Stevens et al, 2020) does this. It states that "The temperature and salinity conditions suggest that, other than the boundary layer regions, water properties conform to Deep Ice Shelf Water (DISW), possibly sourced from Low Salinity Shelf Water (LSSW) or a mixture of high and low salinity shelf water and AASW".

This study indicates that at borehole HWD2, the near-seafloor water is related to this source water but that other regional water masses are not present. Therefore the majority of the cavity is filled with what appears to be different categories of cavity-influenced properties, roughly falling into the envelope of "Deep ISW".

We have clarified this in the text (lines 122-128)

Line 112: “Nutrient concentrations beneath the center of the RIS are...” should be modified to the past tense, when referring to data in this paper. Please check that this is corrected throughout the paper (I note another instance at line 114).

We have corrected this accordingly.

Line 125 – 128: The argument that ice-shelf basal melt contributes ammonium would be strengthened if average concentrations for the Ross Sea were also reported for context.

We have added these values for clarification. For a more comprehensive overview, we have also added nutrient values reported beneath borehole J9. These showed a similar ammonium profile, albeit with higher nutrient concentrations overall. (lines 141-145)

The references (17 and 18) also note that sea ice is a source of ammonium, however, the ice here is glacial ice, so may not be directly comparable. Please address the expected differences between the types.

As the reviewer hints, there is insufficient data on nutrient measurements within ice sheets, especially within hundred-meter-thick ice shelves, to make appropriate comparisons with free-floating sea ice. We have re-written this paragraph to clearly differentiate between 1) observations from shelf melt water, 2) conclusions draw from sea-ice and 3) potential sources of ammonium enrichments in ice shelves (lines 383-407).

Further, ref 13 provides melt rates and other data that should make it possible to calculate what the ammonium concentration of the melt water would have to be in order to account for the observed ammonium enrichment. Such a calculation would make the conclusion seem more plausible, if it supported the idea that melt could supply sufficient ammonium.

We agree that the calculation proposed would be relevant if there were a point source of ammonium at borehole HWD2, from where melting rates are reported, but we cannot determine if this is the case. As we now state in the re-written discussion, even if the IBL is enriched in ammonium with respect to the other water masses, with the current dataset, we cannot discern from an in situ or external source of ammonium. We have discussed the potential sources for the

observed ammonium gradient in the water column, and admit it is still an intriguing phenomenon (lines 383-407).

Along the same lines – how is the ammonium getting into the bottom of the ice shelf? I believe there is a rationale for the high ammonium concentrations in sea ice, but does something like this exist for glacier ice? I am not sure that there is a convincing reason to assume that ice originating from the Antarctic Ice Sheets should contain significant ammonium.

As mentioned above, we have now proposed possible sources of ammonium in shelf ice, based on observations from sea-ice studies. These include atmospheric deposition and organic matter remineralization. Sea-ice is more accessible and has been studied in greater detail (with respect to chemical composition) than Antarctic ice shelves, for which data on chemical composition is very scarce. We have also discussed in the revised text that the enrichment in ammonium in the IBL could be locally sourced (from melting of ice or inglacial debris) but also could be sourced elsewhere (e.g., from sediments at the grounding zone) and transported by melt water beneath the shelf.

As the reviewer suggests, data on glacier ice could serve as an additional proxy to infer the chemical composition of basal ice in ice shelves. However, research on this area has been largely focused on chemistry of glacier melt in inland glaciers. Glacier melt water (from moulins, or glacier mills) has typically a higher proportion of dissolved organic nitrogen (DON) concentrations relative to snow and ice (Wadham et al, 2016). Both DON and ammonium concentrations in glacier runoff are also often elevated during subglacial out-burst events, rising to up to concentrations up to 6 μM (e.g. (Hood et al, 2009). Together with high DON/DOC ratios observed in runoff (Hood & Scott, 2008), the dissolved organic component of N is considered to be acquired via organic matter remineralization by microorganisms on the glacier surface and at the bed (Thomas & Dieckmann, 2002).

While this is still a plausible explanation as to how ammonium reaches the bottom of the ice bed, remineralization of organic matter can also release an important amount of dissolved organic phosphorus, an observation that does not coincide with the phosphate profile beneath the RIS. Further, the potential influence of regional geology in the composition of glacier ice (or glacier ice melt) also begs caution when drawing comparisons.

Given the insufficient evidence to draw convincing conclusions about the sources of ammonium in the IBL, we have resolved to state these uncertainties in the text, and to highlight the scarcity of data with respect to chemical composition of polar ice sheets.

Line 133: I do not believe the acronym “PHP” was previously described. Please spell out on first use.

The acronym was introduced with first use of prokaryotic heterotrophic production; above in the same paragraph (now line 156)

“In contrast, prokaryotic heterotrophic production (PHP) ranged from ...”

Line 165: Please provide more information on the indicator species analysis, including how the signature species are derived, and what parameters were used in the analysis (for example, I believe you can select whether to limit the analysis to only species that represent individual sites, or to allow groups of sites, and this changes the calculations). Where signature species determined for all of the open and sub-ice communities? It is not clear what the signature species noted are signatures of.

The majority of statistical analyses described were performed between two environments: deep (> 200 m) open ocean polar environments, and below-RIS oceanic cavity. These two groups

formed a distinct cluster with respect to other marine communities from the global miTags comparison (Figure 2c, 2d).

We agree that this information was not clearly presented and have specified the groups compared in the text (lines 196-198):

“Indicator Species Analysis (IndVal) congruently identified ‘signature species’ of the ocean cavity (with respect to open mesopelagic polar waters) belonging to the phyla ...”

*The methods section now includes more details on the groups and the parameters used for IndVal analysis. We also specify that the function *multipatt* was used to calculate IndVals, a function that “... looks for indicator species of both individual site groups and combinations of site groups (De Cáceres et al, 2010).” (line 686-687).*

All statistical methods used should also be described in the “Methods” section.

Due to space constraints, we had described all statistical methods in the Supplementary Methods. We have moved them back to the Main Text and added details on the statistical approaches (lines 659-713).

Line 168: If the signature species represent 9% of the RIS community, how do they compare to the open ocean samples?

We have corrected this value, since the average of the samples is closer to 10%, reaching 17% in the deeper layers of the cavity (lines 201-203).

In comparison to deep (meso- and bathypelagic) polar samples, the signature species represent a percentage that varies from 0 to 2% (n=42, average 0.75, stdev 0.55).

These values range from 0-0.6% in polar, surface samples (n=62, average=0.11, stdev= 0.15) and from 0.2-2% in bathypelagic, non-polar environments (n=42, average=0.92, stdev=0.50)

Based on the IndVal index, a particular species can be considered an indicator of the RIS cavity if it appears exclusively in this environment (even though, not necessarily in all samples), or because it is found in all samples beneath the RIS and is largely (but not completely) restricted to this environment. Thus, it is not surprising that the relative abundance of all indicator species is higher in the environment that they are signatures of.

Line 191: In consideration of the general readership of this journal (rather than microbial specialists), please identify whether organisms are members of the Bacteria or Archaea (e.g. Nitrososphaerales).

The text now reads:

“...and relative abundance of the archaeal order Nitrososphaerales”

Also, I note that reference 3 also found abundant Nitrosopumilus and putatively ammonia-oxidizing archaea in deep water. If the current study referenced known water masses, it might be possible to compare these data (or other relevant data) more directly.

The data suggest that the system is not a simple one of a range of external water masses penetrating the cavity in different layers and different regions. Instead, there is (at the observation location) an apparent “source” layer (albeit not strongly identifying with a single known water mass) and then three layers above of varying structure that very likely evolved from this source. Other regional water masses are not present beneath the cavity, which makes data comparisons difficult.

Similarly, direct comparisons of regional studies are not possible due to insufficient data available. Reference 3 (Vick-Majors et al, 2016) described two water masses (HSSW, AASW), but

only relative abundance of Nitrosopumilus for the deeper layer (HSSW). The relative abundances of Nitrosopumilus reported by Vick-Majors and colleagues are 4x lower than in the IBL under the RIS (i.e., ~ 3.5% vs ~13%). In a different comment to this reviewer (see below), we describe the possible factors regulating abundance of ammonium oxidizing archaea in the water column. In particular, seasonal variations in the light regime of the Southern Ocean are likely to have a higher influence in the abundance of Nitrosopumilus and other ammonium oxidizing archaea than the water mass of origin.

We have included reports of similar increases in ammonium oxidizing archaea in the austral winter for comparison (lines 359-368).

Line 200-202: The sentence, “In line with an autotrophic nitrifying bacteria, we identified the determinants of ammonium- or nitrite-dependent carbon fixation via the archaeal 4-hydroxybutyrate cycle and Nitrospina reductive tricarboxylic acid cycle” is confusing. It is not clear how an archaeal C fixation pathway is “in line” with autotrophic nitrifying bacteria. Please consider rewording to make the point clear.

This was a mistake in the text. We have reworded as follows:

“In line with an autotrophic lifestyle, we identified ...”

Line 203: It would be helpful to remind the reader how far beneath the ice-water interface 400 m is.

As mentioned above, we have modified the description of the sampling depths (30m, 180m, and 330m), as distance from ice-water interface. The shallowest sample is 30m from the interface.

Line 265: The statement at the beginning of the discussion implies that the samples collected from this single borehole represent the entire RIS microbiome. I suggest rephrasing such that the inference is more reflective of the scope of the samples reported here.

We have rephrased as follows:

“Collectively, our results provide a first insight on the ecological strategies adopted by communities living in the world’s most extensive sub-ice shelf system.”

Line 266: delete “uniquely”

This has been deleted.

Line 273: What is meant by “net nutrient consumption”? Nutrients are not consumed, and their use does not result in automatic removal from the system – nutrients are transformed and may enter different parts of a system, depending on how they are transformed. Therefore, they have to go somewhere. Where are they going, and in what form, in this system, and how does that explain lower concentrations farther under the ice shelf?

We have removed this sentence from the Discussion.

Line 286-287: The coherence between dark C-fixation and organic carbon demand suggests that it is possible that the ecosystem beneath the RIS is largely sustained by dark C-fixation, but does not indicate that with certainty. A more complete evaluation of this argument would require discussion of organic carbon concentrations and potential sources of DOC in the sub-ice environment. Further, the metagenomic data reported suggest that organisms in the deep waters are focused on breaking down recalcitrant material, not freshly produced, labile DOC. A more complete evaluation should also include this in the discussion.

Following the reviewer’s suggestion, we have commented on possible sources of organic matter beneath the cavity but unfortunately, there is not enough data available (neither in this study nor in the literature) to infer DOC concentrations in this environment. We also discuss the potential of phytoplankton-derived organic matter to contribute to the organic carbon pool, but conclude that locally-sourced DOC is more likely to have a greater importance in this system. (lines 421-443).

Line 295: “Antarctic aquatic environments” implies consideration of both freshwater and seawater (with good reason – it is indeed interesting that this process seems to be important under ice in general); if this is the intention, I would suggest citing a freshwater paper that also shows this such as work from Subglacial Lake Whillans (e.g. Achberger, A. M., B. Christner, A. B. Michaud, J. C. Priscu, M. L. Skidmore, and T. J. Vick-Majors. 2016. Microbial community structure of Subglacial Lake Whillans, West Antarctica. *Front. Microbiol.* 7: 1–13. doi:10.3389/fmicb.2016.01457 or Vick-Majors, T. J., A. C. Mitchell, A. M. Achberger, and others. 2016. Physiological ecology of microorganisms in Subglacial Lake Whillans. *Front. Microbiol.* 7: 1–16. doi:10.3389/fmicb.2016.01705)

Alternatively, if you prefer to maintain a marine focus, you could change “aquatic” to “marine”.

We appreciate the suggestion to include freshwater environments to the manuscript’s discussion but have opted to maintain its focus to a marine environment and have thus applied the change recommended by the reviewer (now line 353).

Line 296: change “sheets” to “shelves”, as refs 9 and 10 both refer to ice shelves, not sheets.

While reference 9 (now 12. Horrigan et al., 1981) refers to ice shelves, reference 10 (now 42. Priscu et al 1990), refers to sea ice. We have changed this accordingly: “...beneath Antarctic ice shelves and sea ice.” (now line 386)

Line 300: please clarify what is meant by “basal layer”.

We changed this to “ice basal boundary layer”, for consistency with the definition of the water masses throughout the text.

Line 301: The mention of “exceptionally high ammonium concentrations” made me wonder whether the ammonia-oxidizers detected are usually associated with high ammonium concentrations, or low ones? How do they compare to ammonia oxidizers found in lower ammonia sub-ice waters, such as in ref 3, or elsewhere in the region? Inclusion of this point in the discussion could bolster claims about the ammonium concentrations.

Beyond ammonium concentrations, there are other factors determining the distribution and abundance of ammonium oxidizing archaea (AOA) in marine environments, which make this point difficult to address.

In general, in marine environments, AOA do appear to be higher in cell numbers and activity in ammonium-rich waters. For example, AOA cell numbers, and ammonium oxidation rates have been observed to positively correlate with ammonium concentrations in the water column of the Sargasso Sea (Newell et al, 2013). Compared to the oligotrophic ocean, higher ammonium oxidation rates are detected in coastal areas, where ambient ammonium concentrations are typically higher (e.g. (Santoro et al, 2010)).

This general trend has also been observed in the Southern Ocean, where amoA gene abundance and potential nitrification rates were reported to be highest in winter, when “competition with phytoplankton was minimal and ammonium concentrations were the highest.” (Christman et al, 2011).

Other seasonal studies in the Southern Ocean also observed overall higher ammonium oxidation rates in winter compared with summer, but the inferred ammonia regeneration flux in winter was lower than in summer (Tolar et al, 2016b). This suggests that inhibition (by e.g., reactive oxygen species, (Tolar et al, 2016a), may be more important than ammonium supply in controlling Thaumarchaeota production during summer. This factor, and the scarcity of comparable datasets makes it difficult to compare abundance of AOA under different sub-ice systems.

For instance, reference 3 (now 4, (Vick-Majors et al, 2016), described two water masses (HSSW, AASW), both of which had ammonium concentrations (0.5-0.7 μ M) comparable to the highest concentrations beneath the RIS (0.4 μ M). This is indicated in the manuscript line 140. Despite the similitude in standing ammonium concentrations, the relative abundance detected for the AOA genus Nitrosopumilus are ~40% lower beneath McMurdo ice shelf (i.e, ~ 3.5% in HSSW, as reported by Vick-Majors and colleagues (no data is provided given for AASW), vs ~13% in the IBL under the RIS). Since the study for Vick-Majors took place in the austral summer, we cannot rule out some form of inhibition or competition for ammonium that results in lower abundances of Nitrosopumilus.

Further, the capability of AOA to use alternative substrates for ammonium oxidation (such as cyanate and urea (e.g. (Kitzinger et al, 2018; Palatinszky et al, 2015)) further complicates the drawing of conclusions on the correlation between AOA and ammonium concentrations in the environment

Despite the challenges for comparisons among different studies, we appreciate the comment by the reviewer as very relevant to the discussion. We have also assessed the potential of the ammonium concentrations to support dark carbon fixation (see lines 369-379, Supplementary Discussion, and comment to Reviewer 1).

Line 308: How is “energy limitation” being defined? An abundance of ammonium is described, which would serve as energy source for abundant community members. No organic C data are provided with which to assess potential energy limitation of heterotrophs. I suggest carefully evaluating what is meant by energy limitation, and considering the large body of literature that exists on that topic.

We have modified this and agree that “oligotrophic” is a more correct definition here, since we describe this system as “highly oligotrophic with respect to labile organic matter” in the paragraph above

Line 309: “lithoautotrophic primary producers” – the paper mostly focused on ammonium oxidizers – are others present?

*In the paragraph above, we have specified this by adding
“These [ammonia and nitrite oxidizers] and other inferred facultative chemolithoautotrophs (such as facultative sulfur-oxidizing bacteria)...” (line 355-356)
We hope this sentence should clarify what is meant with the term “lithoautotrophy” in the following paragraphs in the discussion.*

Line 310: delete “slowly”, as it is subjective.

We have deleted it.

Line 311: I assume that “into the shelf” here means “under the ice shelf”. Please reword for clarity.

We have reworded this as “into the ice shelf cavity”.

Line 312: Similarly, please use consistent terminology, e.g. “below ice-shelf”. Note also that much literature (including ref 13, which provides physical data for this site) refers to below ice shelf waters as “cavities”. The authors may wish to consider modifying their terminology in order to remain consistent with the existing literature on these types of systems. Please take note of this throughout the paper, but I especially noticed references to “shelf” throughout the discussion.

In agreement with this remark, we have referred to the environment under the RIS as an oceanic cavity beneath the ice shelf throughout the manuscript.

Line 312: I do not think that reference 9 reported diatoms in sub-ice shelf waters. Horrigan et al found “virtually undetectable chlorophyll-a”. Perhaps you were thinking of Holm-Hansen et al., 1978. Microbial life beneath the Ross Ice Shelf. *Antarct. J. United States* 4: 129–130

which I believe did report a few dinoflagellates and some diatoms at J9.

This was an error in citation; the correct citation for phytoplankton under the shelf is (Azam et al, 1979), which we selected over the suggested reference, since it provides images of the phytoplankton cells observed.

Your ref 3 also reported phototrophs dominated by *P. antarctica* and some diatoms under the McMurdo Ice Shelf, which is also consistent with water advected from the open Ross Sea and estimated a 10 year decomposition time. These support your contention that some photosynthetically produced organic matter might be important under the ice.

We appreciate the suggestion and have added these arguments to the discussion (lines 433-436).

In addition, I would like to point out that the discussion of the eukaryotic community detected was minimal and suggested the presence of mainly heterotrophic eukaryotes (e.g. line 143). The supplementary data presented are not really sufficient for the reader to draw real conclusions about what was detected. So, if an argument about phytoplankton is going to be made, a more complete discussion of the eukaryotes (or at least the potential phototrophs) should be included.

As the reviewer observed, the contribution of photosynthetic clades is negligible to the eukaryotic community. We have included this observation as part of the argument above (line 437-439).

Line 317: “carbon-limitation” cannot be substantiated, as carbon was not measured, nor were potential sources of carbon comprehensively addressed.

We have rephrased to “... and possibly carbon-limited waters”.

Line 320-322: This argument is a stretch – the trace gas supported life was clearly demonstrated, whereas the work shown here (while important and interesting!) did not demonstrate anything about “minimal nutritional requirements”. In fact, the paper argues the opposite: that this is a nitrogen rich oasis fed by ice-melt.

This sentence has been removed.

Line 329-332: Beginning with “Assuming that”, this is only true if ammonium is indeed coming from the ice shelf, which hasn’t been convincingly argued for, and if ammonium oxidation is indeed the most important autotrophic process beneath the ice. I suggest reconsidering this statement. But, I completely agree with the next sentence: baseline data such as these are extremely important.

In the revised version of the manuscript, we have discussed in depth the potential sources of ammonium in the ice-basal layer (please also see comments above), which we hope can justify our arguments in the final paragraphs of the manuscript. We have also modified this particular statement (now lines 444-466), in agreement with the reviewer’s concerns.

Line 337: please state distance from open water in the site description.

This information was included in the main text but has also been incorporated in the methods section.

Line 354-356: A convincing argument for lack of contamination would show the referenced salinity,

temperature, depth profiles. Similarly, the clearest way to demonstrate that contamination was prevented would be to include the control samples in the MDS plot in S3b.

We have added a more accurate description of the drilling water characteristics (line 489)

We have also included an ordination plot to the Figure S2 including the control samples, as suggested, to demonstrate lack of contamination.

Ca Line 361 onwards: regarding the liquid water that was removed from the WTS for determination of inorganic nutrient concentrations: were controls run to show that this method did not contaminate the samples with, for example, ammonium? Ammonium is easily contaminated. Perhaps this is not a problem, but given the arguments made for high ammonium concentrations, contamination prevention methods should be clearly detailed.

We specify in the methods the use of acid-cleaned samples bottles and rinsing material, but have now also added specifically that the WTS was cleaned after each deployment to avoid cross-sample contamination. For example:

“Before and after deployment, the filter holder was thoroughly cleaned to avoid sample cross-contamination. The pump head interior was also flushed after every deployment with fresh water to prevent salt crystal formation and sample contamination.” (lines 494-496)

“Water samples (150-300 mL) were also collected at the same three depths using the McLane WTS-LV-Bore Hole pump without a filter-holder, to further minimize contamination. Once the pump was brought up, it was run in reverse to collect the water, but excluding the first 30-60 mL of water (used for rinsing).” (lines 506-509)

Further, high ammonium concentrations were detected only in the three replicate samples from the shallowest depth. Given that all samples were treated equally, if this were the result of ammonium contamination in the field, it is to be expected that high concentrations of ammonium would have been detected in all other samples, and this is not the case. Finally, we have also added the following to the methods: “Measurements of nutrient concentrations were routinely corrected with reference blank solutions in each sample run. No anomalies were detected in the blanks, indicative of no source detectable contamination during the measurements.” (lines 538-541)

We also have compared our results to previous observations beneath borehole J9, which show a similar gradient in ammonium concentrations (and reported overall concentrations 10x higher than those measured in this study). Based on past observations and measures taken during sampling, we can confidently argue that the ammonium concentrations measured are not a result of sample contamination.

Line 375-379: These samples should be shown in S3b so that their relationship to the water samples can be easily seen.

As suggested, we have included an ordination plot to the Figure S2 including the control samples, to demonstrate that contamination was prevented.

Line 405: PHP samples are not usually stored frozen because freeze-thaw can damage cells, causing loss of incorporated radioactivity, and leading to diminished detected rates. Please address this.

The leucine method to estimate PHP is based on radioactive leucine incorporation into protein. To effectively concentrate protein samples, trichloroacetic acid (TCA) is usually added to terminate the incubation and induce the precipitation of proteins. Further, TCA precipitation denatures the proteins. Therefore, once TCA is added to the samples, samples can be stored frozen with a negligible effect on the overall radioactivity measured. Freeze-storing samples

previously treated with TCA is a common practice in marine oceanographical measurements of PHP.

Line 413: Please add a citation for the activation energy used.

The value for activation energy was empirically determined for bacterial heterotrophic production in the mesopelagic ocean (Lønborg et al, 2016). We have added the citation to the Methods section (now line 565).

Line 385-388: This seems in contrast with above description of water sample collection for inorganic chemistry from the WTS. Please clarify.

We have modified the Method Section to resolve this inconsistency (lines 506-527).

References:

- Apprill A, McNally S, Parsons R & Weber L (2015) Minor revision to V4 region SSU rRNA 806R gene primer greatly increases detection of SAR11 bacterioplankton. *Aquat Microb Ecol* 75: 129–137
- Aristegui J, Gasol JM, Duarte CM & Herndl GJ (2009) Microbial oceanography of the dark ocean's pelagic realm. *Limnol Oceanogr* 54: 1501–1529
- Azam F, Beers JR, Campbell L, Carlucci AF, Holm-Hansen O, Reid FMH & Karl DM (1979) Occurrence and metabolic activity of organisms under the Ross Ice Shelf, Antarctica, at station J9. *Science* (80-) 203: 451–453
- Baltar F, Aristegui J, Sintés E, Aken HM Van, Gasol JM & Herndl GJ (2009) Prokaryotic extracellular enzymatic activity in relation to biomass production and respiration in the meso- and bathypelagic waters of the (sub)tropical Atlantic. *Environ Microbiol* 11: 1998–2014
- Barria C, Malecki M & Arraiano CM (2013) Bacterial adaptation to cold. *Microbiology* 159: 2437–2443
- Berg C, Vandieken V, Thamdrup B & Jürgens K (2015) Significance of archaeal nitrification in hypoxic waters of the Baltic Sea. *ISME J* 9: 1319–1332
- Brown JH, Gillooly JF, Allen AP, Savage VM & West GB (2004) Toward a metabolic theory of ecology. In *Ecology* pp 1771–1789. John Wiley & Sons, Ltd
- De Cáceres M, Legendre P & Moretti M (2010) Improving indicator species analysis by combining groups of sites. *Oikos* 119: 1674–1684
- Callbeck CM, Lavik G, Ferdelman TG, Fuchs B, Gruber-Vodicka HR, Hach PF, Littmann S, Schoffelen NJ, Kalvelage T, Thomsen S, *et al* (2018) Oxygen minimum zone cryptic sulfur cycling sustained by offshore transport of key sulfur oxidizing bacteria. *Nat Commun* 2018 9: 1–11
- Christman GD, Cottrell MT, Popp BN, Gier E & Kirchman DL (2011) Abundance, diversity, and activity of ammonia-oxidizing prokaryotes in the coastal arctic ocean in summer and winter. *Appl Environ Microbiol* 77: 2026–2034
- Ducklow H (2000) Bacterial production and biomass in the oceans. *Microb Ecol Ocean* ISBN 0-471: 85–120 [PREPRINT]
- Ermolenko DN & Makhatadze GI (2002) Bacterial cold-shock proteins. *Cell Mol Life Sci C* 2002 5911 59: 1902–1913
- Gifford SM, Sharma S, Booth M & Moran MA (2013) Expression patterns reveal niche diversification in a marine microbial assemblage. *ISME J* 7: 281–298
- Grzymiski JJ, Riesenfeld CS, Williams TJ, Dussaq AM, Ducklow H, Erickson M, Cavicchioli R & Murray AE (2012) A metagenomic assessment of winter and summer bacterioplankton from Antarctica Peninsula coastal surface waters. *ISME J* 2012 610 6: 1901–1915
- Hood E, Fellman J, Spencer RGM, Hernes PJ, Edwards R, Damore D & Scott D (2009) Glaciers as a source of ancient and labile organic matter to the marine environment. *Nature* 462: 1044–1047
- Hood E & Scott D (2008) Riverine organic matter and nutrients in southeast Alaska affected by glacial coverage. *Nat Geosci* 2008 19 1: 583–587
- Horrigan SG (1981) Primary production under the Ross Ice Shelf, Antarctica1. *Limnol Oceanogr* 26: 378–382
- Kirchman D, K'nees E & Hodson R (1985) Leucine incorporation and its potential as a measure of protein synthesis by bacteria in natural aquatic systems. *Appl Environ Microbiol* 49: 599–607
- Kitzinger K, Padilla CC, Marchant HK, Hach PF, Herbold CW, Kidane AT, Könneke M, Littmann S, Mooshammer M, Niggemann J, *et al* (2018) Cyanate and urea are substrates for nitrification by Thaumarchaeota in the marine environment. *Nat Microbiol* 2018 42 4: 234–243
- Kumar S, Suyal DC, Yadav A, Shouche Y & Goel R (2020) Psychrophilic *Pseudomonas helmanticensis* proteome under simulated cold stress. *Cell Stress Chaperones* 2020 256 25: 1025–1032

- Landry Z, Swan BK, Herndl GJ, Stepanauskas R & Giovannoni SJ (2017) SAR202 genomes from the dark ocean predict pathways for the oxidation of recalcitrant dissolved organic matter. *MBio* 8
- Lauro FM, McDougald D, Thomas T, Williams TJ, Egan S, Rice S, DeMaere MZ, Ting L, Ertan H, Johnson J, *et al* (2009) The genomic basis of trophic strategy in marine bacteria. *Proc Natl Acad Sci* 106: 15527–15533
- Lønborg C, Cuevas LA, Reinthaler T, Herndl GJ, Gasol JM, Morán XAG, Bates NR & Álvarez-Salgado XA (2016) Depth Dependent Relationships between Temperature and Ocean Heterotrophic Prokaryotic Production. *Front Mar Sci* 3: 90
- López-Urrutia Á & Morán XAG (2007) Resource limitation of bacterial production distorts the temperature dependence of oceanic carbon cycling. *Ecology* 88: 817–822
- Marshall KT & Morris RM (2012) Isolation of an aerobic sulfur oxidizer from the SUP05/Arctic96BD-19 clade. *ISME J* 2013 72 7: 452–455
- Moran MA & Durham BP (2019) Sulfur metabolites in the pelagic ocean. *Nat Rev Microbiol* 2019 1711 17: 665–678
- Newell SE, Fawcett SE & Ward BB (2013) Depth distribution of ammonia oxidation rates and ammonia-oxidizer community composition in the Sargasso Sea. *Limnol Oceanogr* 58: 1491–1500
- Palatinszky M, Herbold C, Jehmlich N, Pogoda M, Han P, Bergen M von, Lagkouvardos I, Karst SM, Galushko A, Koch H, *et al* (2015) Cyanate as an energy source for nitrifiers. *Nat* 2015 5247563 524: 105–108
- Parada AE, Needham DM & Fuhrman JA (2016) Every base matters: assessing small subunit rRNA primers for marine microbiomes with mock communities, time series and global field samples. *Environ Microbiol* 18: 1403–1414
- Phadtare S (2012) Escherichia coli cold-shock gene profiles in response to over-expression/deletion of CsdA, RNase R and PNPase and relevance to low-temperature RNA metabolism. *Genes to Cells* 17: 850–874
- Polz MF, Hunt DE, Preheim SP & Weinreich DM (2006) Patterns and mechanisms of genetic and phenotypic differentiation in marine microbes. *Philos Trans R Soc B Biol Sci* 361: 2009–2021
- Raymond-Bouchard I & Whyte LG (2017) From Transcriptomes to Metatranscriptomes: Cold Adaptation and Active Metabolisms of Psychrophiles from Cold Environments. *Psychrophiles From Biodivers to Biotechnol Second Ed*: 437–457
- Santoro AE, Casciotti KL & Francis CA (2010) Activity, abundance and diversity of nitrifying archaea and bacteria in the central California Current. *Environ Microbiol* 12: 1989–2006
- Shah V, Zhao X, Lundeen RA, Ingalls AE, Nicastro D & Morris RM (2019) Morphological plasticity in a sulfur-oxidizing marine bacterium from the SUP05 clade enhances dark carbon fixation. *MBio* 10
- Smith DC & Azam F (1992) A simple, economical method for measuring bacterial protein synthesis rates in seawater using 3H-leucine 1. *Mar Microb Food Webs* 6: 107–114
- Stevens C, Hulbe C, Brewer M, Stewart C, Robinson N, Ohneiser C & Jendersie S (2020) Ocean mixing and heat transport processes observed under the Ross Ice Shelf control its basal melting. *Proc Natl Acad Sci U S A* 117: 16799–16804
- Swan BK, Martinez-Garcia M, Preston CM, Sczyrba A, Woyke T, Lamy D, Reinthaler T, Poulton NJ, Masland EDP, Gomez ML, *et al* (2011) Potential for chemolithoautotrophy among ubiquitous bacteria lineages in the dark ocean. *Science* (80-) 333: 1296–1300
- Thomas DN & Dieckmann GS (2002) Antarctic Sea Ice--a Habitat for Extremophiles. *Science* (80-) 295: 641–644
- Thompson LR, Sanders JG, McDonald D, Amir A, Ladau J, Locey KJ, Prill RJ, Tripathi A, Gibbons SM, Ackermann G, *et al* (2017) A communal catalogue reveals Earth's multiscale microbial diversity. *Nat* 2017 5517681 551: 457–463
- Tolar BB, Powers LC, Miller WL, Wallsgrove NJ, Popp BN & Hollibaugh JT (2016a) Ammonia Oxidation in the Ocean Can Be Inhibited by Nanomolar Concentrations of Hydrogen Peroxide.

- Tolar BB, Ross MJ, Wallsgrove NJ, Liu Q, Aluwihare LI, Popp BN & Hollibaugh JT (2016b) Contribution of ammonia oxidation to chemoautotrophy in Antarctic coastal waters. *ISME J* 10: 2605–2619
- Tremblay J, Singh K, Fern A, Kirton ES, He S, Woyke T, Lee J, Chen F, Dangl JL & Tringe SG (2015) Primer and platform effects on 16S rRNA tag sequencing. *Front Microbiol* 6: 771
- Vick-Majors TJ, Achberger A, Santibáñez P, Dore JE, Hodson T, Michaud AB, Christner BC, Mikucki J, Skidmore ML, Powell R, *et al* (2016) Biogeochemistry and microbial diversity in the marine cavity beneath the McMurdo Ice Shelf, Antarctica. *Limnol Oceanogr* 61: 572–586
- Wadham JL, Hawkings J, Telling J, Chandler D, Alcock J, O'Donnell E, Kaur P, Bagshaw E, Tranter M, Tedstone A, *et al* (2016) Sources, cycling and export of nitrogen on the Greenland Ice Sheet. *Biogeosciences* 13: 6339–6352
- Williams TJ, Long E, Evans F, DeMaere MZ, Lauro FM, Raftery MJ, Ducklow H, Grzymiski JJ, Murray AE & Cavicchioli R (2012) A metaproteomic assessment of winter and summer bacterioplankton from Antarctic Peninsula coastal surface waters. *ISME J* 2012 610 6: 1883–1900
- Yool A, Martin AP, Fernández C & Clark DR (2007) The significance of nitrification for oceanic new production. *Nat* 2007 4477147 447: 999–1002
- Zhang W, Cao S, Ding W, Wang M, Fan S, Yang B, Mcminn A, Wang M, Xie B & Qin Q-L (2020) Structure and function of the Arctic and Antarctic marine microbiota as revealed by metagenomics. *Microbiome* 8: 1–12

REVIEWERS' COMMENTS

Reviewer #1 (Remarks to the Author):

Thank you for the close attention paid to the previous round of reviews. I have no further suggestions for improving the manuscript.

Reviewer #2 (Remarks to the Author):

I greatly appreciate the effort the authors put into revising the manuscript according to the comments of the reviewers. I also do understand that the sampling point is not the easiest to access in the world and detailed data collection can be problematic. I would also like to note that for example, it is also very difficult to go to the middle of the south Pacific Gyre and difficulty of access to a sampling location cannot compensate for the lack of robust data. Furthermore, as a reviewer I have to consider the robustness, reproducibility and the quality of the presented data and also the new insights that can be extracted from the presented data compared to earlier studies.

In this case the paper of Horrigan from 1981 is an excellent study and when I compare the data presented in this manuscript, the only new (and very valuable) insights can be derived from molecular data. The authors have performed genome/transcriptome and proteome analyses and do indeed build the main message of their manuscript on information gathered from omics. They have performed experiments as much as they could with the limited number of samples the authors have. I do not believe however, that the molecular data is strong enough to back the statements the authors are making especially in the context of carbon fixation. Based on DNA or protein amino acid sequence information it is impossible to predict which direction a reaction can be carried out for bidirectional reactions. Further, in many cases, the authors are only able to detect single or a few genes of very complex biochemical pathways. Consequently, it is very difficult to draw robust conclusions based on the presented omics data.

The authors address my comment on the yield of ammonia-oxidizing microorganisms by a number of values from literature and further calculations. I am aware of the literature data and I am sure the authors have calculated the values correctly. The point I tried to bring across in my criticism is much more fundamental than that. For example, the authors took their biomass yield from Berg et al., where growth of ammonia oxidizing bacteria are measured. The microorganisms grown in the aforementioned study were grown at 22 degrees Celsius with all the metabolic needs of these microorganisms supplied. On the other hand, the authors' sampling location is one of the coldest seawater environments and the microorganisms are under nutrient limitation. The yield and metabolic transformation rates of microorganisms at the aforementioned laboratory conditions cannot be taken and just applied to the environment the authors are working on. This is a very incorrect manner of applying microbial physiology kinetics parameters.

I do understand that "heterotrophic production" is used as a term in marine microbial ecology, but this manuscript is not intended for a marine microbial ecology journal, but a wide audience journal. Therefore, the authors should at least add "growth of heterotrophic microorganisms" in parentheses after "heterotrophic production".

Reviewer #3 (Remarks to the Author):

The authors of Lifting the lid: Phylogenetically and functionally diverse microorganisms reside under the Ross Ice Shelf have done a nice job revising the manuscript and addressing all of my comments. Notably, the improvements to Figure 3 help with interpretation of some of the key findings and I find it much more intuitive. The authors have also removed key discussion of depth-related differences and the metaproteomic data, both of which are changes suggested by me and other reviewers. Combined with the many other revisions, the manuscript now has a sharper focus, is more accurate in its conclusions and I think it will be a nice contribution to the literature.

Reviewer #4 (Remarks to the Author):

I appreciate the author's attention to my comments on the manuscript. In its current form, I believe the conclusions are now better supported by the rest of the text, however, the attribution of ammonia source to the ice shelf is still not well-supported.

Specific Comments:

The next three comments focus on the abstract. Having now read the full revision of the manuscript, I think most of the issues in the abstract are actually addressed in the text. The abstract simply needs to be updated so that it is consistent with the current version of the text.

Line 45: Change "nothing" to "little" – the existing datasets, albeit determined with older methodologies, do not amount to "nothing". The previous work on the RIS at J9, and data collected from the region (e.g. McMurdo Ice Shelf), can be acknowledged without detracting from the unique contributions of the current manuscript.

Line 52: The paper does not present evidence for the source of the listed electron donors. "shelf-sourced" should be deleted.

Line 58-59: Chemosynthetically driven ecosystems – this conclusion can only be made if the alternative hypothesis (that organic matter advected from open water, or melted from the ice drive the ecosystem) can be rejected. The data in the paper do not allow this. So, while there is certainly some evidence for the importance of chemosynthetic metabolisms, I am not convinced that the paper provides evidence that they drive the ecosystem. This statement should be presented with some degree of uncertainty.

Line 80: "into the shelf" – change to "cavity" for consistency

Line 107: "transcription of enzymes" should be re-worded to "transcription of genes". Enzyme production requires steps beyond transcription.

Line 232: There are a couple of problems here. First, energy is not "generated", it is "conserved". Second, I assume putting "ATP" in () is meant to explain why the term "generated" is being used, however, ammonia monooxygenase itself does not produce ATP. These problems could be remedied by rephrasing along the lines of "the most highly transcribed genes involved in autotrophic energy conservation pathways".

Line 266: How were they "observed to support a chemoautotrophic or mixotrophic lifestyle"? This is unclear.

Line 337: "predictably highly oligotrophic with respect to labile organic matter" – this may be true, however, this study did not address this. The statement could be changed to be consistent with the work done in this study, by modifying "predictably" to "predicted to be".

Line 370: The use of "Consistently" here implies that this is a trend, and is routinely found by multiple studies, which cannot be substantiated by a single reference (ref 43). Perhaps what the authors mean is "Consistent with this idea".

Line 383-onwards: This added discussion of potential ammonium sources is useful. However, given that the ammonium concentrations in the open Ross Sea (cited in the paragraph starting at line 131) are significantly greater than the concentrations reported here (and near the edge of the RIS), the argument that the ammonium detected must be coming from the ice shelf to be convincing. How can input from the open Ross Sea be ruled out as a potential source? The discussion is incomplete, because it does not include the Ross Sea data or J9 data that are mentioned earlier in the manuscript.

Line 386: I do not think that ref 44 reported ammonium concentrations.

Line 390: What debris was sequenced?

“Phylogenetically and functionally diverse microorganisms reside under the Ross Ice Shelf” - - point-by-point response to reviewers’ comments

We thank all reviewers for their contribution to peer review this manuscript.

Reviewer #1 (Remarks to the Author):

Thank you for the close attention paid to the previous round of reviews. I have no further suggestions for improving the manuscript.

Reviewer #2 (Remarks to the Author):

I greatly appreciate the effort the authors put into revising the manuscript according to the comments of the reviewers. I also do understand that the sampling point is not the easiest to access in the world and detailed data collection can be problematic. I would also like to note that for example, it is also very difficult to go to the middle of the south Pacific Gyre and difficulty of access to a sampling location cannot compensate for the lack of robust data. Furthermore, as a reviewer I have to consider the robustness, reproducibility and the quality of the presented data and also the new insights that can be extracted from the presented data compared to earlier studies. In this case the paper of Horrigan from 1981 is an excellent study and when I compare the data presented in this manuscript, the only new (and very valuable) insights can be derived from molecular data. The authors have performed genome/transcriptome and proteome analyses and do indeed build the main message of their manuscript on information gathered from omics. They have performed experiments as much as they could with the limited number of samples the authors have.

I do not believe however, that the molecular data is strong enough to back the statements the authors are making especially in the context of carbon fixation.

The data presented in this study shows the genomic potential of the most abundant and transcriptionally active community members in this system. As such, we emphasise throughout the manuscript the nature of omics data as a hypothesis-generation tool. Our conclusions are thus phrased as possibilities and hypothesis worth exploring in the future. It should be noted that, especially with respect to carbon fixation, we base our conclusions on other studies where metabolic rates had been measured but where omics data was missing. We are careful on how these conclusions are phrased, but these are ultimately intended to corroborate or refute previously-made hypothesis from below-shelf systems.

Based on DNA or protein amino acid sequence information it is impossible to predict which direction a reaction can be carried out for bidirectional reactions.

*The majority of the marker genes that we describe in this manuscript have been described to catalyse unidirectional reactions. In the cases for enzymes with ambiguous directionality (such as *r-dsrA*), we have inferred the direction of the reaction by multiple sequence comparison of previously characterized enzymes (**Fig. S12**). These are based on the biology of known cultivated dissimilatory sulphate-reducing micro-organisms.*

For this particular reaction (the reverse, or oxidative Dsr-pathway), it was generally assumed that versions of the same enzymes used in the reductive pathway could operate in the reverse direction. However, differences between these two enzyme types, and these can be observed in phylogenetic analyses of DsrA and DsrB proteins. As mentioned to the comment above, our predictions made from the omics data offer a means to infer the most probable metabolism taking place in this system.

Therefore, in this last revision of the document, we have been careful not to overstate the conclusions made in this respect.

Further, in many cases, the authors are only able to detect single or a few genes of very complex biochemical pathways. Consequently, it is very difficult to draw robust conclusions based on the presented omics data.

This study is based on a large dataset of inferred gene functions, which is derived from all of the identified genes (or, open reading frames) in the metagenomic assemblies and the single-cell assembled genomes.

To be able to systematically identify the presence or absence of a particular pathway in this large dataset, our approach was based on the identification of marker genes each pathway. These marker genes are essential and thus always present in the metabolic pathway (e.g., RubisCO in the Calvin-Benson cycle). Since our aim was to describe the potential of the community to perform certain metabolic reactions, the use of marker genes fulfilled our needs, and justifies the simplification of the dataset. Inclusion in the analysis of other genes encoding enzymes involved in a particular pathway would have resulted in more confusing results (e.g., some enzymes are present in multiple metabolic pathways, others are not essential in all organisms), and would have not provided more information related to the presence/absence of a metabolic trait.

The authors address my comment on the yield of ammonia-oxidizing microorganisms by a number of values from literature and further calculations. I am aware of the literature data and I am sure the authors have calculated the values correctly. The point I tried to bring across in my criticism is much more fundamental than that. For example, the authors took their biomass yield from Berg et al., where growth of ammonia oxidizing bacteria are measured. The microorganisms grown in the aforementioned study were grown at 22 degrees Celsius with all the metabolic needs of these microorganisms supplied. On the other hand, the authors' sampling location is one of the coldest seawater environments and the microorganisms are under nutrient limitation. The yield and metabolic transformation rates of microorganisms at the aforementioned laboratory conditions cannot be taken and just applied to the environment the authors are working on. This is a very incorrect manner of applying microbial physiology kinetics parameters.

We agree that the physiological conditions of the organisms used in the study from Berg et al might be very different to those found under the ice shelf. Whether the archaea beneath the Ross Ice shelf are nutrient limited is something difficult to assess, in particular given the large range of substrate affinities for known AOAs. In principle, based on the data of cultured representatives the concentrations of ammonia are sufficient for this reaction to take place unconstrained.

Unfortunately, growth yields or substrate kinetic analyses are usually done in optimal growth conditions, and we do not have other references to derive our calculations from. Therefore, we have now specifically stated that the growth efficiency data is derived from an optimally-growing culture and acknowledged that our estimates should be considered "an upper end for autotrophy fueled by ammonium oxidizing archaea".

I do understand that "heterotrophic production" is used as a term in marine microbial ecology, but this manuscript is not intended for a marine microbial ecology journal, but a wide audience journal. Therefore, the authors should at least add "growth of heterotrophic microorganisms" in parentheses after "heterotrophic production".

We have amended this to the main text:

"...prokaryotic heterotrophic production (PHP, i.e growth of heterotrophic organisms) ..."

Reviewer #3 (Remarks to the Author):

The authors of Lifting the lid: Phylogenetically and functionally diverse microorganisms reside under the Ross Ice Shelf have done a nice job revising the manuscript and addressing all of my comments. Notably, the improvements to Figure 3 help with interpretation of some of the key findings and I find it much more intuitive. The authors have also removed key discussion of depth-related differences and the metaproteomic data, both of which are changes suggested by me and other reviewers. Combined with the many other revisions, the manuscript now has a sharper focus, is more accurate in its conclusions and I think it will be a nice contribution to the literature.

Reviewer #4 (Remarks to the Author):

I appreciate the author's attention to my comments on the manuscript. In its current form, I believe the conclusions are now better supported by the rest of the text, however, the attribution of ammonia source to the ice shelf is still not well-supported.

Specific Comments:

The next three comments focus on the abstract. Having now read the full revision of the manuscript, I think most of the issues in the abstract are actually addressed in the text. The abstract simply needs to be updated so that it is consistent with the current version of the text.

Line 45: Change “nothing” to “little” – the existing datasets, albeit determined with older methodologies, do not amount to “nothing”. The previous work on the RIS at J9, and data collected from the region (e.g. McMurdo Ice Shelf), can be acknowledged without detracting from the unique contributions of the current manuscript.

This has been changed

Line 52: The paper does not present evidence for the source of the listed electron donors. “shelf-sourced” should be deleted.

This is now deleted

Line 58-59: Chemosynthetically driven ecosystems – this conclusion can only be made if the alternative hypothesis (that organic matter advected from open water, or melted from the ice drive the ecosystem) can be rejected. The data in the paper do not allow this. So, while there is certainly some evidence for the importance of chemosynthetic metabolisms, I am not convinced that the paper provides evidence that they drive the ecosystem. This statement should be presented with some degree of uncertainty.

The text reads now: “and support the hypothesis that ocean cavity waters are primarily chemosynthetically-driven systems.”

Line 80: “into the shelf” – change to “cavity” for consistency

This is now changed

Line 107: “transcription of enzymes” should be re-worded to “transcription of genes”. Enzyme production requires steps beyond transcription.

Agreed, this has been reworded accordingly

Line 232: There are a couple of problems here. First, energy is not “generated”, it is “conserved”.

Second, I assume putting “ATP” in () is meant to explain why the term “generated” is being used, however, ammonia monooxygenase itself does not produce ATP. These problems could be remedied by rephrasing along the lines of “the most highly transcribed genes involved in autotrophic energy conservation pathways”.

We agree with the reviewer’s suggestion and have changed the text accordingly

Line 266: How were they “observed to support a chemoautotrophic or mixotrophic lifestyle”? This is unclear.

We have rephrased to

“Many members of the microbial community are capable of supporting their growth or persistence beneath the shelf via a chemoautotrophic or mixotrophic lifestyle”

Line 337: “predictably highly oligotrophic with respect to labile organic matter” – this may be true, however, this study did not address this. The statement could be changed to be consistent with the work done in this study, by modifying “predictably” to “predicted to be”.

We have changed to “predicted to be”

Line 370: The use of “Consistently” here implies that this is a trend, and is routinely found by multiple studies, which cannot be substantiated by a single reference (ref 43). Perhaps what the authors mean is “Consistent with this idea”.

In agreement with the suggestion, we have modified the text.

Line 383-onwards: This added discussion of potential ammonium sources is useful. However, given that the ammonium concentrations in the open Ross Sea (cited in the paragraph starting at line 131) are significantly greater than the concentrations reported here (and near the edge of the RIS), the argument that the ammonium detected must be coming from the ice shelf to be convincing. How can input from the open Ross Sea be ruled out as a potential source? The discussion is incomplete, because it does not include the Ross Sea data or J9 data that are mentioned earlier in the manuscript.

To complete the discussion, we have added a new paragraph to this section arguing against the possibility that the ammonium profile originates from the open Ross Sea, based on the current circulation model under the Ross Ice Shelf. We have also emphasised the ambiguity of the origin of ammonium into the water column in the revised Figure 6.

In relation to this point, we have also modified the header for the Results Section that previously read: “Ice melting of the basal layer influences nutrient profiles...” to “The water column under the Ross Ice Shelf is characterized by a steep vertical ammonium gradient” (l 111)

Line 386: I do not think that ref 44 reported ammonium concentrations.

The reviewer is right; the reference reports ammonium concentrations, but these are not part of a peer reviewed study. Ref 44 (Priscu et al., Mar. Ecol. Prog. 1990) reads the following:

“...levels of up to 5 μ M have been measured in melt ice from the lower 5 cm of 2 m long ice cores in the land-fast ice of McMurdo Sound (J. C. Priscu unpubl., Sullivan & Buck unpubl.). “

Thus, we have omitted the reference from the discussion.

Line 390: What debris was sequenced?

We have now specified “englacial debris”, which is mentioned above, in the text.

The Methods section has also been amended and reads the following:

“...and sediments dislodged from the ice shelf (identified as englacial debris) and collected with the reaming tool.”